# Basal IFNλ2/3 signaling is required for ISG expression and viral control in human intestinal epithelial cells

**Yagmur Keser, Zehra Sena Bumin, Amelia Perez Valiente, Sorin O. Jacobs, Steeve Boulant\*, Megan L. Stanifer**[ID]\*

Department of Molecular Genetics and Microbiology, University of Florida, College of Medicine, Gainesville, Florida, United States of America

\* m.stanifer@ufl.edu (MLS); s.boulant@ufl.edu (SB)

## Abstract

Interferon-lambdas (IFNλs) serve as critical mediators of antiviral defense at mucosal surfaces. Beyond their established role in regulating innate immune responses during infection, recent evidence demonstrates that IFNλs are constitutively expressed in pathogen-free environments, termed "basal" IFN expression. While intestinal epithelial cells constitutively express all basal IFNλ subtypes (IFNλ1, IFNλ2, and IFNλ3), their individual contributions to antiviral immunity remain poorly defined. Here, we systematically investigate the distinct roles of IFNλ1 and IFNλ2/3 in regulating intrinsic antiviral immunity using human intestinal epithelial T84 cells. Through genetic depletion of IFNλ1 or IFNλ2/3, we show that basal IFNλ2/3, but not IFNλ1, is essential for restricting replication and spread of diverse viruses, including vesicular stomatitis virus (VSV), mammalian orthoreovirus (MRV), rotavirus (RV), and vaccinia virus (VV). Transcriptomic profiling revealed that IFNλ2/3 selectively controls the basal expression of interferon-stimulated genes (ISGs), including key antiviral effectors and components of the IFN signaling machinery (e.g., STAT1, STAT2, IRF9). Loss of IFNλ2/3 reduced total STAT1 protein levels and blunted responsiveness to exogenous IFNλ, indicating compromised interferon signaling capacity. Furthermore, basal IFNλ2/3 was required for activating paracrine JAK/STAT signaling and ISG induction in neighboring bystander cells, thereby amplifying antiviral protection across the epithelial layer. These findings reveal a functional hierarchy among IFNλ subtypes and establish IFNλ2/3 as the dominant, non-redundant regulators of epithelial immune readiness. Our study provides the first comprehensive analysis of basal IFNλ subtype functions in the gut epithelium and underscores the central role of basal IFNλ2/3 in maintaining mucosal antiviral defense.

**Data availability statement:** The RNA-seq data generated in this study have been deposited in the NCBI Gene Expression Omnibus (GEO) under accession number GSE296527.

**Funding:** This work was supported by the National Institute of Allergy and Infectious Diseases (NIAID) of the National Institutes of Health under award number 1R01AI185510 to SB and 1R01AI189780 to MLS. YK, ZSB and MLS were financially supported by 1R01AI189780. SOJ and SB and financially supported by 1R01AI185510. The funders had no role in study design, data collection and analysis, decision to publish, or preparation of the manuscript.

**Competing interests:** The authors have declared that no competing interests exist.

## Author summary

Interferon-lambdas (IFNλs) are antiviral molecules that help protect the surfaces of our body, such as the gut and lungs, from infection. While IFNλs are best known for being produced during viral infections, recent work has shown that some IFNλs are also made at low levels even when no pathogen is present. This "basal" IFN activity acts like a constant security system that keeps cells alert and ready to respond quickly when a virus arrives. However, the specific roles of the different flavors of IFNλ (i.e. IFNλ1, IFNλ2, and IFNλ3) in this baseline protection have remained unclear.

In this study, we investigated how each type of IFNλ (IFNλ1, IFNλ2, and IFNλ3) contributes to antiviral readiness in human intestinal epithelial cells. By selectively removing each IFNλ, we discovered that IFNλ2 and IFNλ3, but not IFNλ1, are essential for maintaining this built-in antiviral state. Cells lacking IFNλ2/3 became highly vulnerable to a wide range of viruses and lost the ability to activate key antiviral genes. Our findings reveal a previously unrecognized hierarchy among IFNλs and highlight IFNλ2/3 as critical guardians of gut antiviral defense, even before infection occurs.

## Introduction

Type I and type III interferons (IFNs) are central to the vertebrate innate immune response controlling viral infections [1–3]. Type III IFNs (IFNλs), the most recently identified type of IFNs, share several characteristics with type I IFNs [4,5]. Both families signal through receptor complexes that activate the JAK/STAT pathway, leading to the transcriptional induction of hundreds of interferon-stimulated genes (ISGs), which collectively inhibit viral replication and spread [6,7]. However, a notable distinction between type I and III IFNs, lies in their receptor distribution: while type I IFN receptors (IFNAR1/IFNAR2) are broadly expressed across most cell types [8], the IFNλ receptor (IFNLR) is largely restricted to epithelial cells of the respiratory and gastrointestinal tracts, skin, and a subset of immune cells [9,10]. This restricted expression pattern position IFNλs as specialized guardians of epithelial barriers, where they play a pivotal role in preventing pathogen dissemination and maintaining tissue homeostasis [9,10]. The IFNλ family in humans is comprised of four members: IFNλ1 (IL-29), IFNλ2 (IL-28A), IFNλ3 (IL-28B), and IFNλ4. IFNλ2 and IFNλ3 are highly conserved and widely expressed across mammalian species, sharing 96% amino acid identity [11]. In contrast, IFNλ1 is expressed in humans and a subset of other primates, such as chimpanzees and great apes, but is absent in several other mammals, including mice, where it exists as a non-functional pseudogene [12–14]. IFNλ4 shows notable evolutionary variability across human populations, being expressed only in a subset of individuals. While certain alleles associated with IFNλ4 expression have been negatively selected in some populations, its relatively high

frequency in others, particularly African populations, indicates that IFNλ4 may also have conferred context-dependent advantages [4,5,15]. Although some studies have compared the antiviral activities of different IFNλ subtypes, whether different IFNλ subtypes have distinct biological functions remains unclear [11,16,17].

IFNs are primarily produced by cells in response to viral infections, orchestrating the body's antiviral defense mechanisms. Upon viral entry and replication, cellular pattern recognition receptors (PRRs) detect viral components, initiating a signaling cascade that leads to IFN production. Key PRRs include RIG-I-like receptors (RLRs), Toll-like receptors (TLRs), and cyclic GMP-AMP synthase (cGAS) [18–20]. RLRs, such as RIG-I and MDA5, primarily recognize double-stranded RNA (dsRNA) in the cytoplasm, TLRs sense bacterial membranes and viral genomes, while cGAS senses cytosolic DNA [21–23]. These PRRs initiate downstream signaling cascades through adaptor proteins (MAVS, TRIF/MyD88, or STING), leading to activation of TANK-binding kinase-1 (TBK1), which in turn phosphorylates interferon regulatory factor-3 (IRF3) and interferon regulatory factor-7 (IRF7). The phosphorylation of IRF3 and IRF7 results in their dimerization and translocation into the nucleus, where they act as transcription factors driving the expression of IFNs [24–26]. Once secreted, IFNs bind to their receptors on the same (autocrine) or neighboring (paracrine) cells, triggering the Janus kinase (JAK)-Signal Transducer and Activator of Transcription (STAT) signaling pathway. Phosphorylated STAT1 and STAT2 associate with interferon regulatory factor 9 (IRF9) to form the interferon-stimulated gene factor 3 (ISGF3) complex. ISGF3 acts as a transcription factor, regulating the expression of numerous interferon-stimulated genes (ISGs) that play pivotal roles in inhibiting viral replication and supporting the immune response against infections [8,27].

Traditionally, IFNs have been recognized for their role in driving an antiviral state in the host cells in response to pathogen challenges. However, recent studies have revealed that constitutive, or basal, expression of IFNs occurs even in pathogen-free environments, indicating a role beyond immediate immune defense [28,29]. Notably, type I IFNs, such as IFNβ, are constitutively expressed at low levels in healthy tissues/cells, where they play a crucial role in maintaining immune homeostasis [30–32]. This basal type I IFN expression is essential for regulating the expression of the components of the JAK/STAT signaling pathway, like STAT1 and STAT2, thereby priming cells for a swift and robust response upon encountering pathogens [30]. In the absence of IFNβ signaling, studies have observed reduced levels of STAT1, STAT2, IRF1, and IRF7 in cells maintained in sterile environment, making them less responsive to interferon signaling and more susceptible to subsequent infection [33–35]. This underscores the importance of basal interferon signaling in sustaining the readiness of the innate immune system.

Previously studies aiming at understanding the functions of basal interferon largely focused on type I IFNs [28–35]. More recently, IFNλs have also been shown to contribute to immune homeostasis. Recent studies demonstrated that the intestinal microbiota could drive localized, homeostatic IFNλ production that primes discrete epithelial niches for antiviral protection in mice [36,37]. In parallel, our recent work revealed that IFNλs are also constitutively expressed in human intestinal epithelial cells under sterile conditions, independent of microbial cues [36,38]. Their expression correlates with epithelial confluency and is driven by the detection of cytosolic mitochondrial DNA via the cGAS/STING pathway [39]. These findings collectively highlight that IFNλ signaling operates across multiple homeostatic layers: one induced by the microbiota and one arising intrinsically from self-DNA sensing. Despite the well-established antiviral role of virus-induced IFNλs, the specific contributions of basal IFNλ signaling remains poorly understood. How individual IFNλ subtypes uniquely shape immune preparedness and epithelial homeostasis in human intestinal epithelial cells is still unclear.

Using CRISPR-edited human intestinal epithelial cell lines deficient in IFNλ1 or IFNλ2/3, we demonstrate that basal IFNλ2/3, but not IFNλ1, is essential for restricting infection by a broad range of viral pathogens, including rotavirus (RV), mammalian reovirus (MRV), vesicular stomatitis virus (VSV), and vaccinia virus (VV). Mechanistically, IFNλ2/3 governs both autocrine and paracrine JAK/STAT signaling and is required to sustain basal expression of core ISGs, including STAT1, IRF7, and RIG-I. Loss of basal IFNλ2/3 results in reduced total STAT1 levels and impaired responsiveness to exogenous IFNλ stimulation, demonstrating that basal expression of IFNλ2/3 is critical to protect against forthcoming viral infection by regulating the immune readiness of host cells. These findings uncover a previously unappreciated, non-redundant role for

basal IFNλ2/3 compared to IFNλ1 in sustaining epithelial antiviral defense and establish a functional hierarchy among IFNλ subtypes. Our work not only expands current understanding of mucosal immunity but also lays a foundation for future therapeutic strategies aimed at enhancing epithelial barrier defense through targeted modulation of basal IFNλ2/3 signaling.

## Results

**Intestinal epithelial cells upregulate IFNλ1 and IFNλ2/3 upon virus infection.** The importance of IFNλs in controlling viral infection in intestinal epithelial cells has been well described using cells depleted of the IFNλ receptor [6,37,39–45], studies in mice have further demonstrated that loss of the cytokines Ifnl2 and Ifnl3 phenocopies the absence of Ifnlr signaling, underscoring their essential role in mucosal antiviral defense [44]. However, the relative contribution of the individual human IFNλ subtypes, IFNλ1, IFNλ2, and IFNλ3, remains much less characterized. To investigate their contribution in controlling virus infection, the human intestinal epithelial T84 cells were infected with different viruses. We chose four different model viruses from different families and with different genomes to ensure that the measured contribution of each IFNλ was not virus specific. We employed the negative-sense single-stranded RNA virus vesicular stomatitis virus (VSV) expressing GFP (VSV-GFP), the enteric double-stranded RNA viruses mammalian orthoreovirus (MRV) and rotavirus (RV) encoding the fluorescent protein UnaG (RV-UnaG), and the double-stranded DNA virus vaccinia virus (VV) expressing GFP (VV-GFP). Live-cell fluorescent imaging (VSV-GFP, RV-UnaG, or VV-GFP) or immunostaining of the MRV non-structural protein µNS confirmed that T84 cells are readily infectable by these viruses (Fig 1A–1D). To evaluate the expression and secretion levels of IFNλ1, IFNλ2, and IFNλ3, infected cells and their supernatants were collected at indicated time points. Transcriptional upregulation of IFNλs was assessed by quantitative real-time PCR (qRT-PCR) (Fig 1E–1H) and secretion of IFNλs in the supernatant of infected cells was addressed using enzyme-linked immunosorbent assay ELISA (Fig 1I–1L). Due to their high sequence similarity, IFNλ2 and IFNλ3 were analyzed together using qRT-PCR and ELISA to measure their transcriptional upregulation and secretion, respectively. Infection of the human intestinal epithelial cells by all viruses upregulated both IFNλ1 and IFNλ2/3 at the RNA (Fig 1E–1H) and protein levels (Fig 1I–1L).

## Recombinant IFNλ subtypes exhibit comparable antiviral activity against viruses

To assess whether IFNλ subtypes exhibit differential antiviral activity against these distinct classes of viruses, we pretreated the intestinal epithelial cells T84 cells with increasing concentrations (0.0001–300 ng/mL) of IFNλ1, IFNλ2, or IFNλ3 for 24 hours prior to infection. Cells were subsequently infected with VSV expressing the luciferase gene (VSV-Luc), MRV, RV-UnaG, or VV-GFP in the presence of IFNλs. Viral replication was quantified using luciferase assays for VSV-Luc, immunostaining for the MRV non-structural protein µNS, and live-cell fluorescent imaging for RV-UnaG and VV-GFP (Fig 2A–2D). While at low concentrations of IFNλs, IFNλ2 and IFNλ3 were slightly more antiviral compared to IFNλ1 against MRV (Fig 2B) and RV-UnaG infections (Fig 2C), all IFNλ subtypes conferred a dose-dependent antiviral protection against all tested viruses. Of note, individual viruses displayed variable sensitivity to IFNλ treatment: VSV-Luc was highly sensitive to IFNλs and was fully inhibited at concentrations superior to 10 ng/mL whereas VV-GFP infection was more resistant to IFNλ treatment and could not be fully suppressed, exhibiting only approximately 30% reduction in viral replication at the highest cytokine doses (Fig 2A and 2D). These differences are likely due to the high sensitivity of VSV to IFNs [46–48] and the efficient capacity of VV to block the IFN response [49,50]. Together, these results indicate that treatment of intestinal epithelial cells with recombinant IFNλ1, IFNλ2, and IFNλ3 exhibits broadly similar antiviral activities.

## Genetic depletion of IFNλ2/3, but not IFNλ1, drastically enhances viral replication and spread in human intestinal epithelial cells

While treatment with recombinant IFNλ subtypes revealed comparable antiviral efficacy across diverse viruses (Fig 2), this approach may not fully illustrate the importance of each IFNλ subtype when expressed at the endogenous levels. This might be relevant in T84 human intestinal epithelial cells as IFNλ2/3 appears more expressed compared to IFNλ1 in

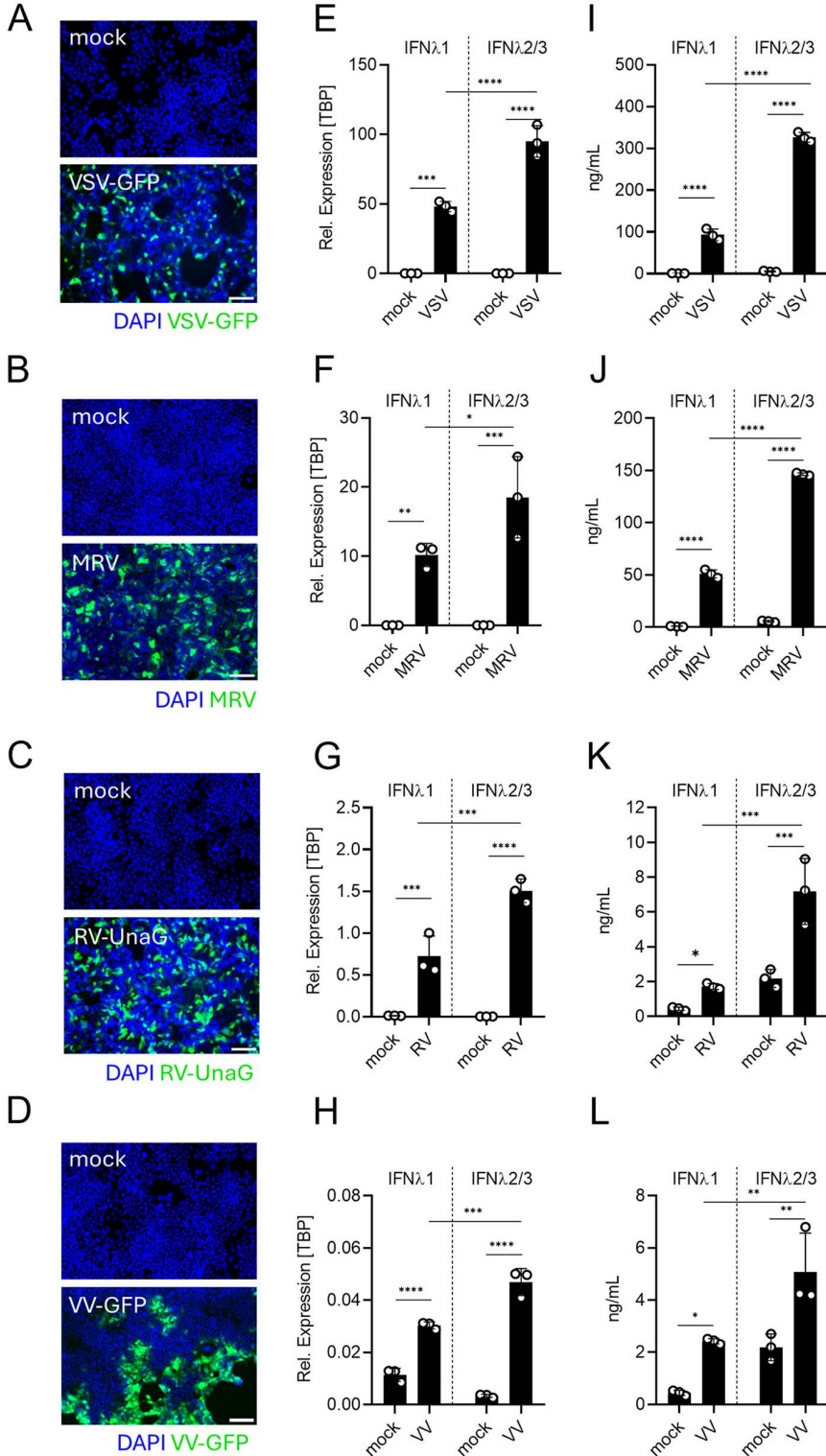

**Fig 1. IFNλ response of human intestinal epithelial cells to different virus types.** (A–D) T84 cells were seeded in 48-well plates and infected two days later with (A) VSV-GFP at an MOI of 1 for 7 hours, (B) MRV at an MOI of 1 for 16 hours, (C) RV-UnaG at an MOI of 1 for 16 hours and (D) VV-GFP at an MOI of 1 for 16 hours. (A) VSV-GFP (C) RV-UnaG and (D) VV-GFP infection was evaluated using live-cell microscopy; nuclei were stained with Hoechst. (B) MRV infection was assessed by immunostaining against the MRV µNS protein, and nuclei was stained using DAPI. (A–D) Representative

fluorescence images showing virus (green) and nuclei (blue). Scale bar = 100 µm. (E–H) Total RNA was extracted from mock-infected or virus-infected T84 cells at (E) 7hpi of VSV-GFP and at 16hpi of (F) MRV, (G) RV-UnaG and (H) VV-GFP, followed by qRT-PCR analysis of IFNλ1 and IFNλ2/3 expression. Gene expression levels were normalized to TBP. (I–L) Supernatants collected from infected T84 cells at (I) 7hpi of VSV-GFP and at 16hpi of (J) MRV, (K) RV-UnaG and (L) VV-GFP, were analyzed by ELISA to quantify secreted IFNλ1 and IFNλ2/3 proteins following infection. Data represent ≥3 independent biological replicates. Statistical significance was determined by unpaired t-test (*P < 0.05, **P < 0.01, ***P < 0.001, ****P < 0.0001). Error bars represent standard deviation with the mean as the center.

response to viral infection (Fig 1E–1L). To dissect the specific contributions of IFNλ1 versus IFNλ2/3 under physiological conditions, we generated T84 cell lines deficient of IFNλ1 (IFNλ1 KO) and IFNλ2/3 (IFNλ2/3 KO) using CRISPR/Cas9 approaches. Generation of IFNλ1 and IFNλ2/3 KO cells was validated by Sanger sequencing (S1A Fig). To functionally validate that these cell lines were knocked out for IFNλ1 and IFNλ2/3, we transfected WT, IFNλ1 KO, and IFNλ2/3 KO cells with poly I:C, a synthetic dsRNA mimic. In WT and IFNλ2/3 KO cells, IFNλ1 secretion was robustly induced following poly I:C transfection, whereas IFNλ1 KO cells showed no detectable secreted IFNλ1 (S1B Fig). Reciprocally, WT and IFNλ1 KO cells exhibited strong IFNλ2/3 secretion in response to poly I:C transfection, while IFNλ2/3 KO cells did not secrete IFNλ2/3 (S1C Fig). To directly address the importance of endogenous IFNλ1 and IFNλ2/3 in controlling viral infection we infected T84 WT, IFNλ1 KO, and IFNλ2/3 KO cells with VSV-GFP, RV-UnaG, MRV, and VV-GFP. Virus infection was assayed by live-cell fluorescent microscopy for VSV-GFP at 7 hpi, for RV-UnaG at 12 hpi, and for VV-GFP at 16 hpi (Fig 3A, 3C, and 3D). Indirect immunofluorescence of the non-structural protein µNS was used to evaluate MRV infection at 16 hpi (Fig 3B). Results showed that T84 cells depleted of IFNλ2/3 were infected to a higher degree as compared to WT and IFNλ1 KO cells for all four viruses tested (Fig 3A–3D). Interestingly, loss of IFNλ1 did not increase virus infection and IFNλ1 KO cells were infected to a similar degree as WT cells (Fig 3A and 3C) or less than WT cells (Fig 3B and 3D). These findings suggest that IFNλ2/3 KO cells are intrinsically more susceptible to viral infection compared to WT and IFNλ1 KO cells. To further investigate whether IFNλ1 and IFNλ2/3 are important to control virus spread over multiple rounds of infection, WT and IFNλ KO cell lines were infected with VSV-GFP, RV-UnaG, and VV-GFP. Virus infections were monitored by live-cell fluorescence microscopy for 48 hours at an imaging interval of 2-hours (S2A–S2C Fig). In agreement with our previous findings, viral spread was markedly enhanced in IFNλ2/3 KO cells as compared to WT cells at both early and late stages of infection for all three viruses (S2A–S2C Fig). In contrast, IFNλ1 KO cells exhibited no differences in viral spread as compared to WT cells across all tested viruses (S2A–S2C Fig).

To verify that the observed phenotypes were not due to clonal artifacts in the IFNλ KO cell lines, we repeated these experiments using polyclonal IFNλ knockout cells (S3 Fig). First, we assessed IFNλ production following poly(I:C) stimulation by ELISA, which confirmed a marked reduction of the corresponding IFNλ subtypes in each polyclonal KO population (S3A–S3C Fig). Next, we tested their susceptibility to viral infection by infecting IFNλ1 KO and IFNλ2/3 KO polyclonal cells with VSV-GFP (S3D, S3E Fig) and RV-UnaG (S3F and S3G Fig). Live-cell fluorescence microscopy revealed that IFNλ2/3 KO polyclonal cells were consistently more susceptible to infection, while IFNλ1 KO polyclonal cells displayed infection levels similar to, or lower than, WT cells (S3D–S3G Fig). Together, these findings highlight the critical role of endogenous IFNλ2/3 in conferring the human intestinal epithelial cell T84 cells antiviral protection. On the contrary, endogenous IFNλ1 does not appear to significantly contribute to this antiviral defense.

## Inhibition of basal IFNλ2/3, not IFNλ1, signaling enhances virus infection

IFNλs are classically recognized for their induction upon viral infection; however, we recently demonstrated that IFNλs are also constitutively expressed in sterile, uninfected epithelial cells [39]. This basal IFNλ expression is driven by cGAS–STING activation in response to cytosolic mitochondrial DNA (mtDNA) [39]. After observing that genetic depletion of IFNλ2/3 markedly increases early viral infection (Fig 3), we next sought to determine whether this heightened susceptibility reflects the loss of basal IFNλ2/3 signaling or impaired virus-induced IFNλ2/3 production. To distinguish between these possibilities, we first

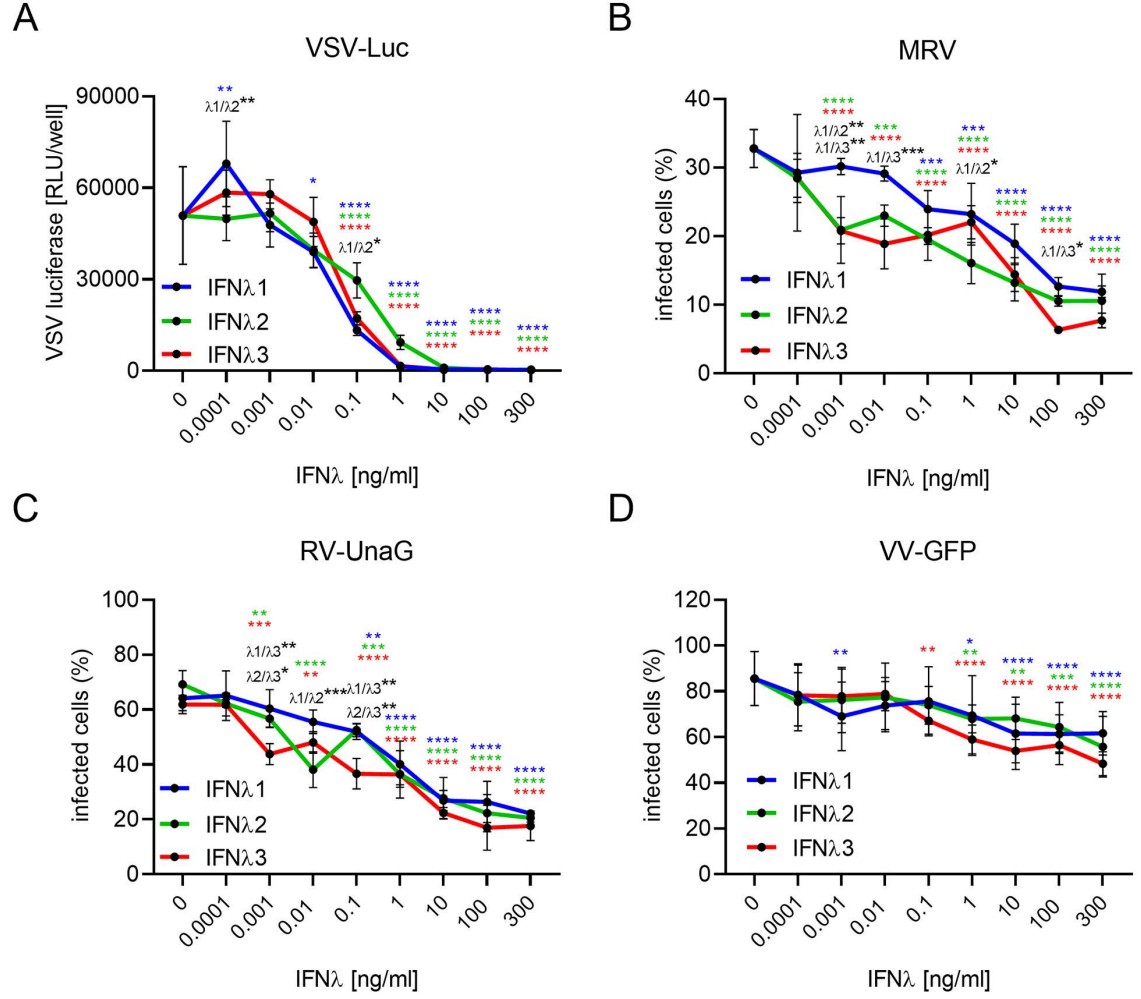

**Fig 2. Recombinant IFNλ subtypes exhibit comparable antiviral activity against diverse virus types in intestinal epithelial cells.** T84 cells were seeded in 96-well plates and treated the following day with increasing concentrations (0.0001–300 ng/mL) of recombinant IFNλ1, IFNλ2, or IFNλ3 for 24 hours prior to infection. Cells were then infected with (A) VSV-Luc, (B) MRV, (C) RV-UnaG, or (D) VV-GFP, each at a multiplicity of infection (MOI) of 1. Infections were maintained in the presence of indicated dose of recombinant IFNλ1, IFNλ2, or IFNλ3. Infections were analyzed 7 hours post-infection (hpi) for VSV-Luc and 16 hpi for MRV, RV-UnaG, and VV-GFP. (A) VSV-Luc infection was quantified by luciferase assay. (B) MRV infection was assessed by immunofluorescence staining against the µNS protein, with DAPI used for nuclear staining. (C, D) RV-UnaG and VV-GFP infections were monitored via live-cell imaging; nuclei were stained with Hoechst. Data represent ≥3 independent biological replicates. Statistical significance between IFNλ-treated conditions and the untreated control (0 ng/ml) was determined using two-way ANOVA with Sidak's post hoc correction (*$P < 0.05$, **$P < 0.01$, ***$P < 0.001$). Color-coded significance markers indicate comparisons between different doses and 0 ng/mL for each IFNλ subtype (IFNλ = blue, IFNλ2 = green and IFNλ3 = red). If not specified, comparisons are not significant (ns). Error bars represent standard deviation with the mean as the center.

used the STING inhibitor H151 [51] to selectively block basal IFNλ signaling, followed by infection with VSV, whose detection is mediated predominantly by RNA sensors such as RIG-I and TLRs [52]. T84 WT cells were pre-treated with H151 for 2 days prior to VSV-Luc infection, and inhibition of basal IFNλ1 and IFNλ2/3 expression was confirmed by qRT-PCR (Fig 4A and 4B). Importantly, suppressing basal IFNλ signaling did not impair virus-induced IFNλ responses; in fact, H151-treated cells mounted a stronger VSV-induced IFNλ expression than untreated controls (Fig 4A and 4B). Despite intact inducible signaling, inhibition of basal IFNλs significantly increased viral infection in WT cells (Fig 4C). This effect was specifically due

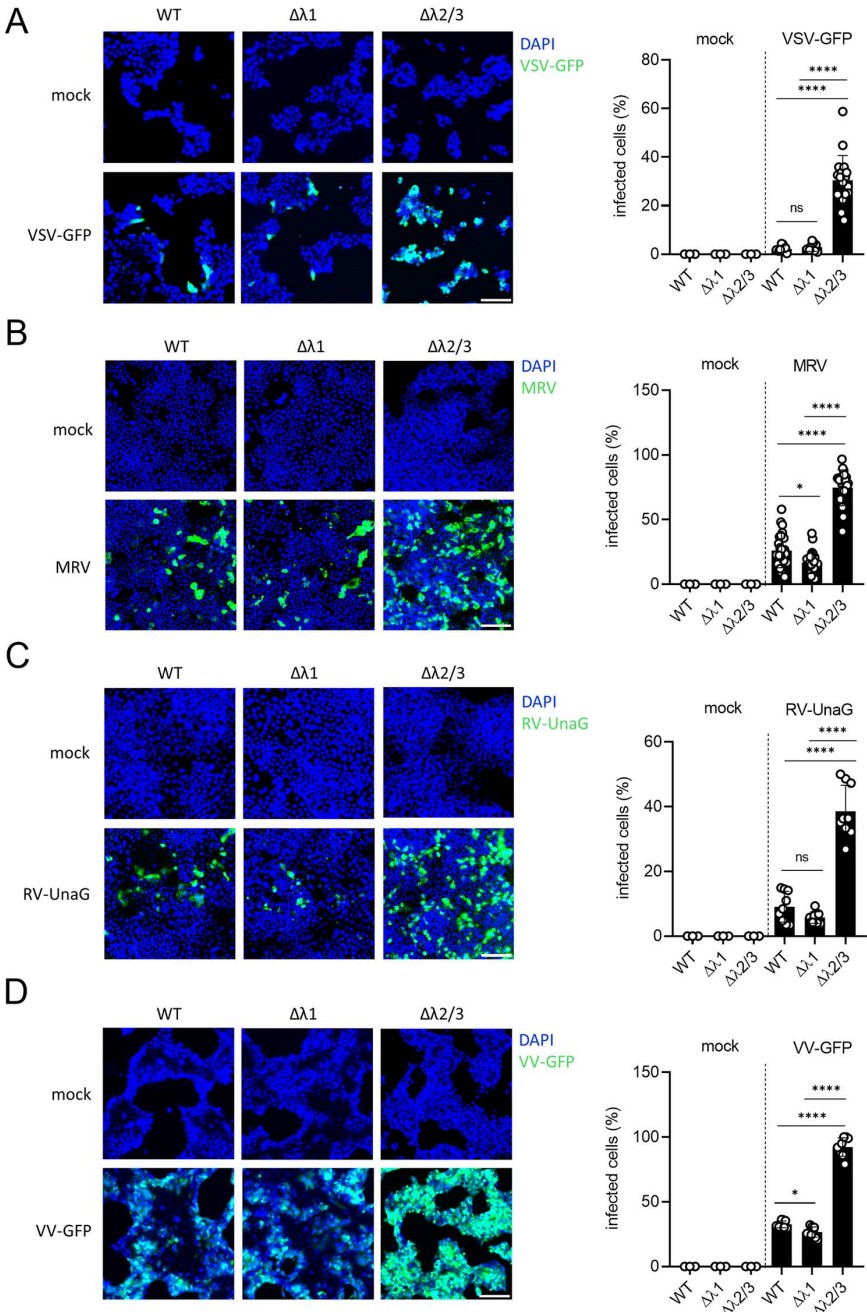

**Fig 3. Genetic depletion of IFNλ2/3, but not IFNλ1, enhances early infection by diverse viruses in human intestinal epithelial cells.** T84 WT, IFNλ1 KO, and IFNλ2/3 KO cells were seeded in 48-well plates and infected the following day. (A) Cells were infected with VSV-GFP (MOI = 1), and infection was assessed at 7 hours post-infection (hpi) by live-cell microscopy. Nuclei were stained with Hoechst (blue), and infected cells are shown in green. (B) Cells were infected with MRV (MOI = 1), and infection was evaluated at 16 hpi by immunostaining against the MRV μNS protein; nuclei were counterstained with DAPI. (C) Cells were infected with RV-UnaG (MOI = 1), and infection was measured by live-cell microscopy at 12 hpi. (D) Cells were infected with VV-GFP (MOI = 1), and infection was evaluated at 16 hpi using live-cell microscopy. (A–D) Representative images (left) and corresponding quantification (right) are shown for each virus. Scale bar = 100 μm. Data represent ≥3 independent biological replicates. Statistical significance was determined by two-way ANOVA (*P < 0.05, ****P < 0.0001, ns = not significant). Error bars represent standard deviation with the mean as the center.

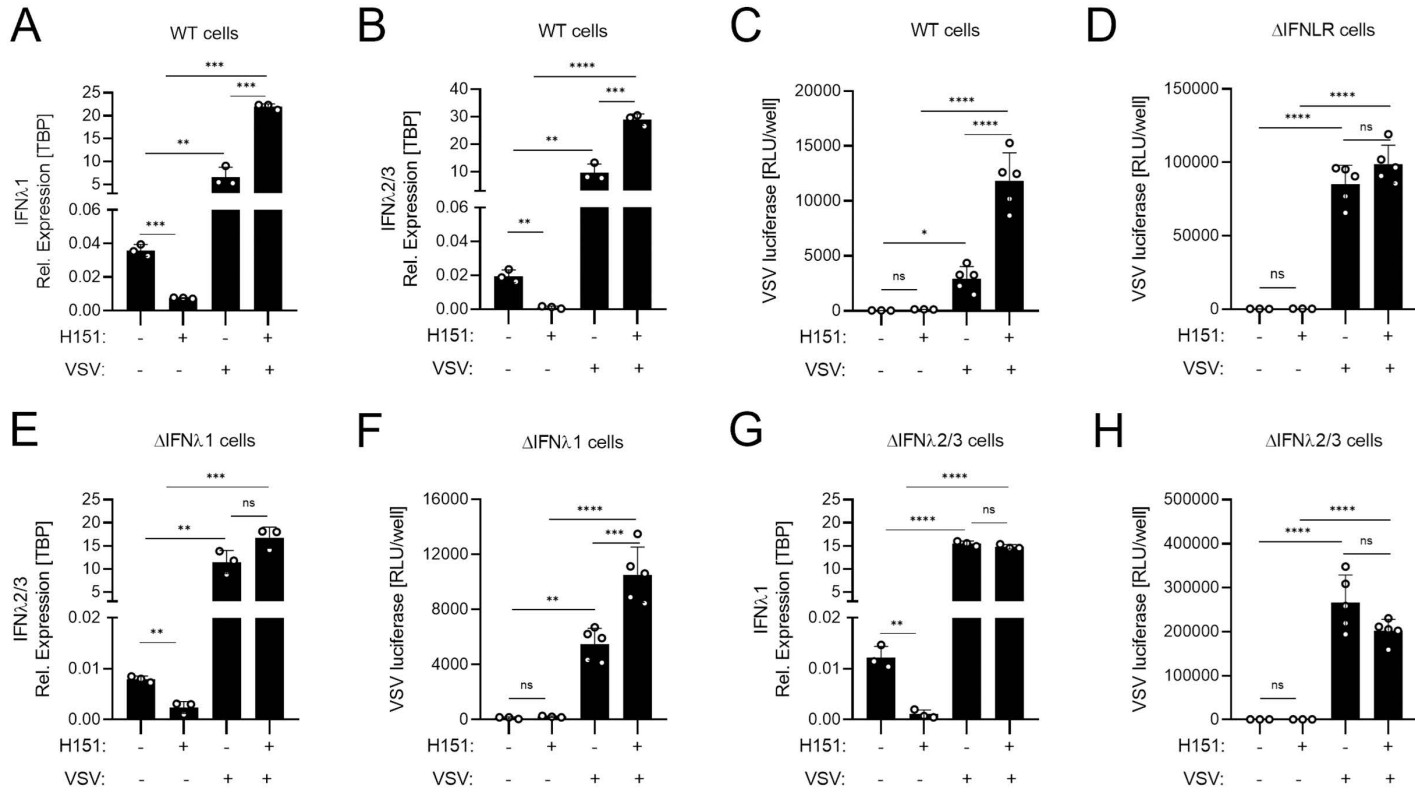

**Fig 4. STING-driven basal IFNλ2/3, not IFNλ1, expression is crucial to maintain basal ISGs and antiviral state.** (A–H) T84 WT, IFNLR KO, IFNλ1 KO, and IFNλ2/3 KO cells were seeded in (A, B, E, G) 48-well plate as 200,000 cell/well or (C, D, F, H) 98-well plate as 50,000 cell/well, and next day the media was replaced with 20 μM H151 (STING inhibitor) or DMSO (solvent control). Cells were incubated with H151 or DMSO for 2 days and subsequently infected with VSV-Luc (MOI = 1) for 7 hours in the continued presence or absence of H151. (A, B, E, G) Basal and virus-induced IFNλ1 and/ or IFNλ2/3 expression was assessed by qRT-PCR. (C, D, F, H) Virus infection was quantified by luciferase assay. Relative expression was normalized to TBP. Data represent n ≥ 3 biological replicates. Statistical significance was determined using one-way ANOVA with multiple comparisons (*P < 0.05; **P < 0.01; ***P < 0.001; ****P < 0.0001; ns = not significant). Error bars represent standard deviation with the mean shown at the center.

to loss of basal IFNλ signaling, as H151 treatment did not substantially alter viral infection in IFNLR KO cells [6] (Fig 4D). Notably, IFNLR KO cells showed a pronounced increase in initial infection of VSV-GFP, RV-UnaG, and VV-GFP infection (S4A–S4D Fig). In contrast, IFNAR KO cells [6] exhibited only a modest increase in VSV-GFP and RV-UnaG infection and showed no enhanced susceptibility to VV-GFP (S4A–S4D Fig). These results highlight that, in T84 epithelial cells, IFNλ signaling provides the dominant antiviral protection, surpassing the contribution of type I IFN signaling.

To further dissect IFNλ subtype-specific contributions to the basal antiviral state, we inhibited basal IFNλ2/3 expression in IFNλ1 KO cells (Fig 4E and 4F) using H151 (Fig 4E and 4F). Virus-induced IFNλ2/3 in IFNλ1 KO cells remained intact after H151 treatment, confirming selective inhibition of the basal, not induced, response (Fig 4F). Notably, inhibition of basal IFNλ2/3 in IFNλ1 KO cells significantly increased viral infection (Fig 4F). Similarly, we inhibited basal IFNλ1 expression in IFNλ2/3 KO cells (Fig 4G and 4H), and virus-induced IFNλ1 in IFNλ2/3 KO cells (Fig 4H). In contrast to WT and IFNλ1 KO cells, inhibition of basal IFNλ1 in IFNλ2/3 KO cells had no impact on infection (Fig 4H). Together, these results demonstrate that STING-driven basal IFNλ2/3, but not IFNλ1, is essential for maintaining epithelial antiviral readiness. More importantly, they reveal a functional hierarchy among IFNλ subtypes at basal levels, with IFNλ2/3 serving as the dominant contributors to the pre-existing antiviral state.

**IFNλ2/3 are the primary regulators of basal ISG expression in intestinal epithelial cells**

Basal IFN expression is known to regulate steady-state ISG levels and plays a critical role in preparing cells for future viral challenges. In the context of type I IFNs, basal IFNβ expression has been shown to sustain the expression of key ISGs that confer intrinsic antiviral protection, while also regulating components of the JAK/STAT signaling pathway, thereby priming cells for a swift and robust response upon encountering pathogens [28–35]. Using H151, we could show that suppressing basal IFNλ2/3, but not IFNλ1, significantly increased viral susceptibility (Fig 4). These findings demonstrate that basal IFNλ2/3 expression is essential for establishing the epithelial antiviral state and suggest a subtype-specific hierarchy in which IFNλ2/3 serves as the dominant contributor to the intrinsic antiviral protection.

To determine whether intrinsic antiviral pathways are differentially regulated in the absence of basal IFNλ signaling, and to define the individual contributions of IFNλ subtypes to this basal antiviral state in intestinal epithelial cells, we performed transcriptomic analysis of T84 WT, IFNλ1 KO, and IFNλ2/3 KO cells. In addition, we also included IFNLR KO cells [6] to serve as a control for investigating the global importance of basal IFNλ in regulating basal ISG expression. Principal component analysis (PCA) revealed that WT and IFNλ1 KO cells clustered together and were clearly separated from IFNLR KO and IFNλ2/3 KO cells (Fig 5A). Compared to WT cells, IFNLR KO cells and IFNλ2/3 KO cells displayed significant downregulation of canonical ISGs (Fig 5B and 5D), while IFNλ1 KO cells exhibited a smaller subset of differentially expressed ISGs with modest fold-change reductions (Fig 5C).

Similarly, Gene Ontology (GO) enrichment analysis of differentially expressed genes (DEGs) from each comparison (WT vs. KO cells) revealed robust enrichment of antiviral and innate immune pathways in WT cells, which were diminished in all knockout conditions (Fig 5E). Notably, loss of IFNλ2/3 or IFNLR resulted in a more substantial reduction in key pathways such as response to virus, regulation of innate immune response, and interferon-mediated signaling as compared to IFNλ1 KO cells (Fig 5E). Examination of the top 25 DEGs associated with the GO term "defense response to other organism" (GO:0098542) revealed marked downregulation of key antiviral ISGs (MX1, OAS1, IFIT1, ISG15) in both IFNλ2/3 KO and IFNLR KO cells (Fig 5F). In contrast, these genes were only modestly reduced in IFNλ1 KO cells (Fig 5F). Moreover, essential components of the IFN signaling machinery, including RIG-I, IRF7, STAT1, STAT2, and IRF9, were significantly reduced in IFNλ2/3 KO and IFNLR KO cells, but not in IFNλ1 KO cells (Fig 5F), further supporting a dominant role for basal IFNλ2/3 in the regulation of these immune processes. To rule out the possibility that transcriptomic changes arose from compromised cell viability rather than loss of IFN signaling, we evaluated gene sets associated with cell survival and apoptosis. Gene set enrichment analysis (GSEA) revealed no significant enrichment of these pathways (S5A–S5C Fig), and cytotoxicity assays confirmed that all cell lines maintained comparable viability (S5D Fig), supporting that the observed differences were specifically due to disruption of IFNλ signaling.

To validate the loss of essential components of the IFN signaling machinery, we quantified ISG transcript and protein levels in T84 WT, IFNλ1 KO, IFNλ2/3 KO, and IFNLR KO cells. ISGs were selected from the top 25 most differentially expressed genes related to the GO term "defense response to other organisms" in the RNA-seq dataset (Fig 5F, asterisk-marked genes). qRT-PCR confirmed modest reductions in MX1, OAS1, ISG15, IRF7, RIG-I, and IFIT1 mRNA levels in IFNλ1 KO cells compared to WT cells. Importantly, a robust reduction of ISG expression was observed in IFNλ2/3 KO and IFNLR KO cells compared to WT cells (Fig 6A). This phenotype was also recapitulated in polyclonal (pc) KO populations: IFNλ1 KO polyclones displayed ISG levels comparable to WT cells, whereas IFNλ2/3 KO polyclones showed a significant reduction in basal ISG expression relative to WT, confirming that these differences reflect genuine subtype-specific functions rather than clonal variation (S6A–S6D Fig). Expression of the housekeeping gene TBP remained stable across all cell lines (S7 Fig), confirming that the observed differences were not due to general transcriptional defects. Consistent with our RNAseq and qRT-PCR results, Western blot analysis showed that IFNλ1 KO cells displayed a modest change in protein expression of MX1, IRF7, RIG-I, ISG15, and STAT1 compared to WT cells, while IFNλ2/3 KO and IFNLR KO cells showed reduced expression of these ISGs at the protein levels compared to WT cells (Fig 6B).

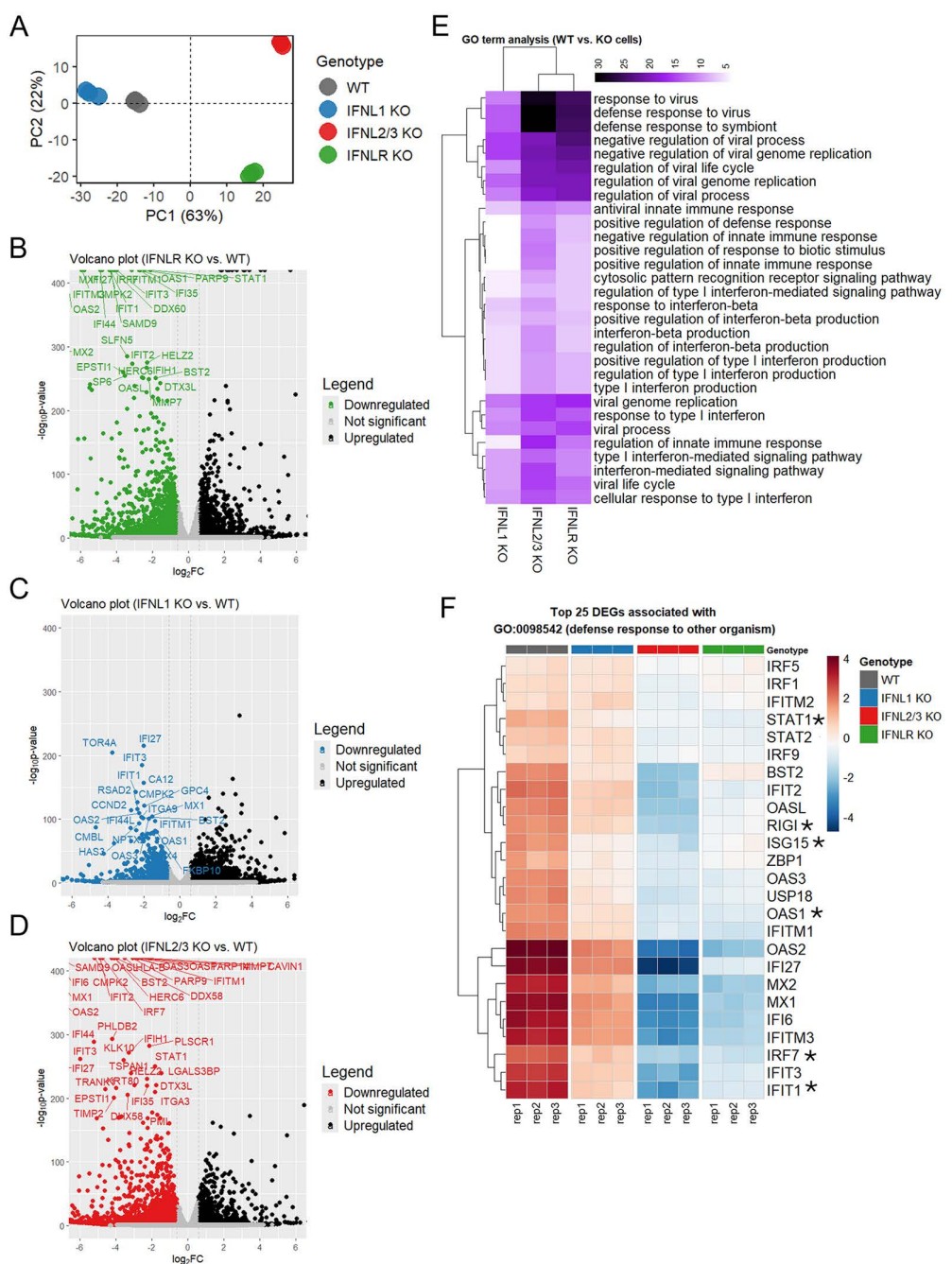

**Fig 5. RNA sequencing reveals the dominant role of basal IFNλ2/3 signaling in maintaining basal ISG levels in intestinal epithelial cells.** T84 WT, IFNλ1 KO, IFNλ2/3 KO, and IFNLR KO cells were seeded in 48-well plates and subjected to RNA sequencing three days post-seeding. (A) Principal Component Analysis (PCA) plot displaying the distribution of T84 WT, IFNλ1 KO, IFNλ2/3 KO, and IFNLR KO cells based on their gene expression profiles. Each point represents an individual sample, colored according to the experimental group. (B) T84 IFNLR KO vs. WT cells, (C) T84 IFNλ1 KO vs. WT cells, (D) T84 IFNλ2/3 KO vs. WT cells. (B-D) Each point represents a gene, plotted by its fold-change (x-axis) and statistical significance (-log10 p-value, y-axis). Genes with significant differential expression ($p<0.05$) are highlighted in black (upregulated) and green, blue and red (downregulated). The most downregulated genes in KO cells are labeled. (E) Gene Ontology (GO) enrichment analysis was performed for Biological Process (BP) terms using the top 500 differentially expressed genes (DEGs) from each WT vs. KO cells comparison. The heatmap displays the top 30 GO terms ranked by their average significance score, and hierarchically clustered based on the similarity of their enrichment profiles. The color intensity represents the statistical significance of each GO term's enrichment, calculated as the $-\log_{10}$(p-value). (F) The heatmap displays the top 25 differentially expressed genes

associated with the biological process "innate immune response" (GO:0045087). Rows represent genes, columns represent samples, and hierarchical clustering was applied to both. Color intensity indicates relative expression levels (red: high; blue: low). Asterisk-marked genes are further validated in Fig 6A and 6B. Data represents three independent biological replicates.

Together, these results demonstrate that IFNλ2/3, rather than IFNλ1, are the primary regulators of basal ISG expression in intestinal epithelial cells. While IFNλ1 contributes modestly to the maintenance of basal innate immune signaling, the loss of IFNλ2/3 recapitulates the full extent of ISG suppression observed in IFNLR-deficient cells, underscoring the predominant and non-redundant role of IFNλ2/3 in maintaining epithelial basal immunity.

### Reduced STAT1 levels in IFNλ2/3 KO cells attenuate responsiveness of cells to exogenous IFNλ stimulation

As basal IFNλ2/3 not only sustain basal ISG expression but also support the basal expression of key components of the interferon signaling machinery, such as STAT1, STAT2, and IRF9 (Figs 5F and 6A, B), we next sought to determine whether IFNλ2/3 KO cells retain the ability to respond to interferon stimulation at levels comparable to WT cells. To assess this, we treated T84 WT, IFNλ1 KO, and IFNλ2/3 KO cells with recombinant IFNλ1–3 and evaluated JAK/STAT pathway activation by Western blot for total and phosphorylated STAT1 (Fig 6C), along with ISG expression by qRT-PCR (Fig 6D). In both WT and IFNλ1 KO cells, IFNλ treatment induced robust STAT1 phosphorylation and strong upregulation of ISGs (Mx1, OAS2, ISG15 and IFIT1) (Fig 6C and 6D). In contrast, although IFNλ2/3 KO cells remained responsive to stimulation, they displayed markedly reduced STAT1 phosphorylation and lower ISG induction, correlating with decreased total STAT1 protein levels (Fig 6C, 6D). These results indicate that basal IFNλ2/3 signaling is essential for maintaining expression of core signaling components, such as STAT1, thereby priming intestinal epithelial cells for a robust response to interferon. Altogether, our findings underscore the critical role of basal IFNλ2/3 in establishing and maintaining the responsiveness of intestinal epithelial cells to interferons.

### Basal IFNλ2/3, not IFNλ1, is crucial to induce basal ISGs expression

Our transcriptomic analysis suggests that IFNλ2/3 are the dominant contributors to the induction of basal ISGs in intestinal epithelial cells (Figs 5 and 6). To assess whether basal IFNλs can initiate paracrine JAK/STAT signaling, we performed a supernatant transfer assay. Cell culture supernatants (referred to as conditioned media) of T84 WT, IFNλ1 KO, and IFNλ2/3 KO cells were collected. As a negative control, we included conditioned media from IRF3 KO cells (S8A Fig), given that IRF3 is the essential transcription factor for IFNλ production. As expected, IRF3 KO cells failed to produce either IFNλ1 or IFNλ2/3 under basal conditions (S8B Fig) or upon viral infection (S8C–S8D Fig). The collected conditioned media were immediately applied to naïve WT T84 cells, and STAT1 phosphorylation was assessed at 1 hpt as a readout of JAK/STAT activation (Fig 7A and 7B). Conditioned media from WT cells induced robust STAT1 phosphorylation, whereas media from IRF3 KO cells failed to activate STAT1, displaying levels comparable to culture media alone (DMEM-F12) (Fig 7B, left panel). This confirms that IRF3-dependent basal IFN production is required to activate JAK/STAT signaling (Fig 7B, left panel). Cells treated with conditioned media from IFNλ1 KO cells displayed a modest reduction in STAT1 phosphorylation relative to WT-conditioned media (Fig 7B, left panel), consistent with a partial contribution of IFNλ1 to basal signaling. In striking contrast, conditioned media from IFNλ2/3 KO cells almost completely lost the ability to induce STAT1 phosphorylation, demonstrating that basal IFNλ2/3 are the primary drivers of paracrine basal IFN signaling (Fig 7B, left panel). To confirm that STAT1 activation was specifically mediated by IFNλs present in the conditioned media, IFNLR KO cells were treated in parallel. As expected, no STAT1 phosphorylation was observed in IFNLR KO cells following any conditioned media treatment (Fig 7B, right panel), establishing that IFNλs, and not other secreted factors, are responsible for the observed JAK/STAT activation.

To determine whether the basal levels of IFNλs contained in the conditioned media is sufficient to upregulate ISGs, we extended the supernatant transfer assay by treating recipient cells with conditioned media for 24 hours and quantifying

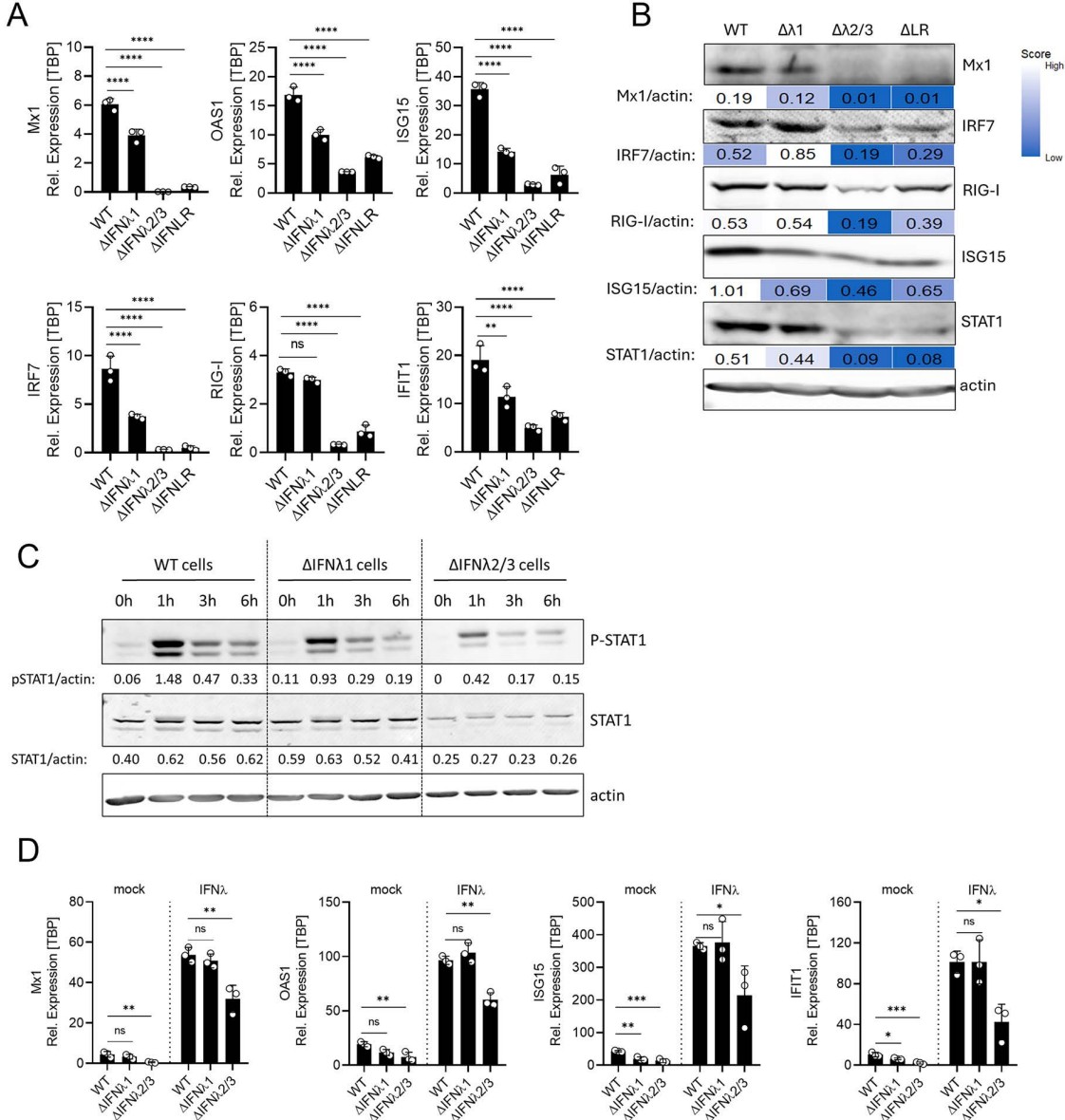

**Fig 6. The loss of basal IFNλ2/3 signaling strongly inhibits key ISGs expression and sensitivity to interferon stimulation.** (A) qRT-PCR analysis of select ISGs Mx1, OAS1, ISG15, IRF7, RIG-I, and IFIT1 in T84 WT, IFNλ1 KO, IFNλ2/3 KO, and IFNLR KO three days post-seeding. Relative expression was normalized to TBP. (B) Western blot analysis of select ISGs (Mx1, IRF7, RIG-I, ISG15 and STAT1) in T84 WT, IFNλ1 KO, IFNλ2/3 KO, and IFNLR KO three days post-seeding. Mx1, IRF7, RIG-I, ISG15 and STAT1 protein abundance was quantified relative to actin as loading control. Representative images shown. (C) T84 WT, IFNλ1 KO, IFNλ2/3 KO cells were treated with recombinant IFNl1-3 proteins (100ng/mL) and cells were collected at 0-, 1-, 3-, and 6-hours post-treatment. Western Blot analysis of p-STAT1 and STAT1 was performed. P-STAT1 and STAT1 abundances were quantified relative to actin as loading control. Representative images shown. (D) Same as (C) but ISG (Mx1, OAS1, ISG15 and IFIT1) induction was assessed by qRT-PCR 24 h post-treatment. Relative expression was normalized to TBP. Data represent n ≥ 3 biological replicates. Statistical significance was determined using two-way ANOVA (*P < 0.05, $P < 0.01$ **, $P < 0.001$ ***, $P < 0.0001$ ****, ns = not significant). Error bars represent standard deviation with the mean as the center.

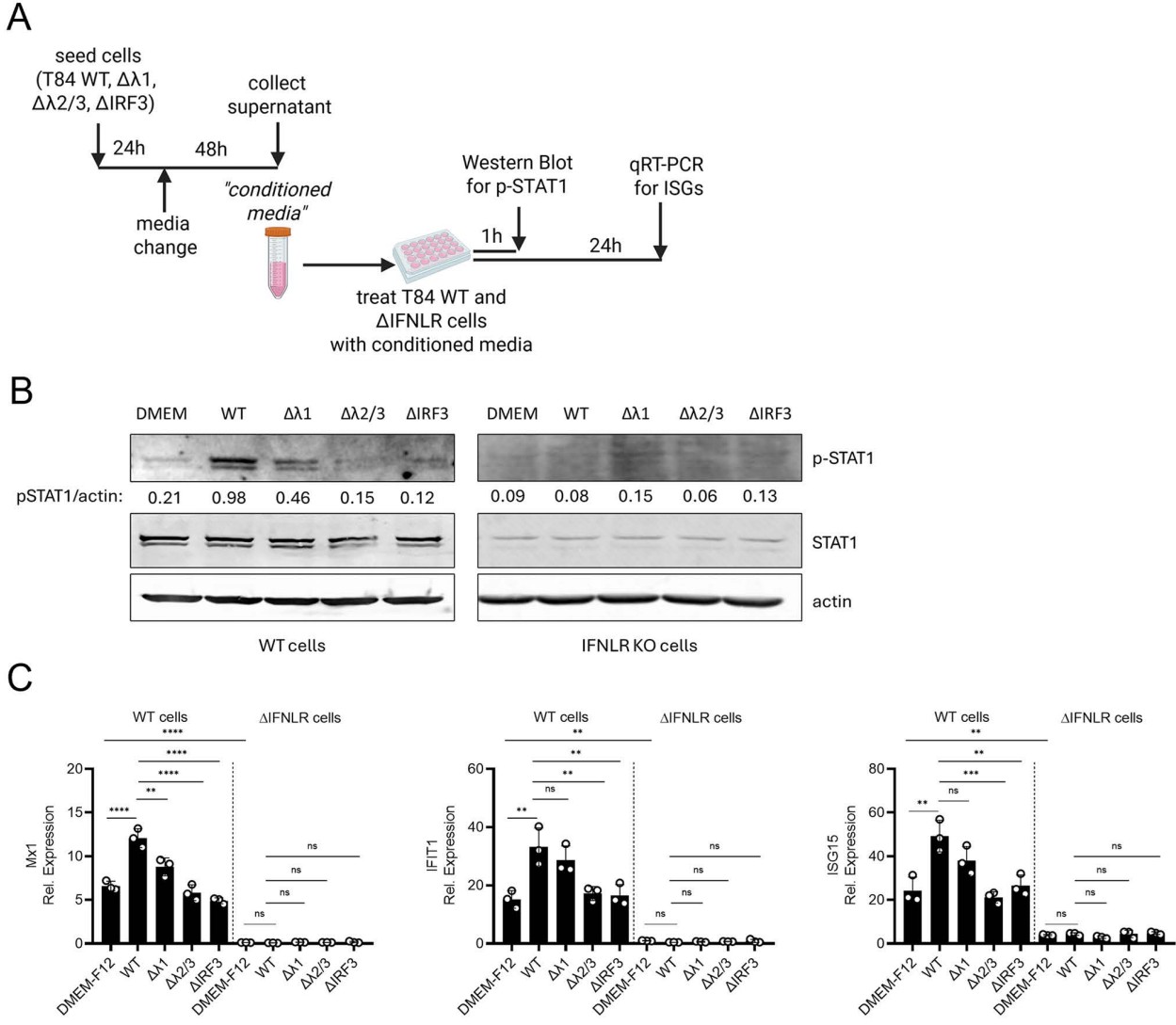

**Fig 7. Basal IFNλ2/3 is the primary driver of paracrine JAK/STAT signaling and ISG induction in human intestinal epithelial cells.** (A–C) T84 WT, IFNλ1 KO, IFNλ2/3 KO, and IRF3 KO cells were seeded in 6 well plates as 2x10^6 cells/well, and the media was changed the following day with 1.5 mL fresh media. Two days later, the cell supernatant was collected after centrifugation at 2000rpm for 5 minutes (referred as conditioned media), and used to treat T84 WT and IFNLR KO cells. Cells were treated with culture media (DMEM-F12) as control. (A) Schematic representation of experimental design was created in BioRender Keser,Y. (2025) https://BioRender.com/6ln3qq4. (B) At 1-hour post-treatment (hpt), cells were harvested for Western blot analysis of STAT1 phosphorylation. P-STAT1 protein abundance was quantified relative to total actin, loading control. Representative images shown. (C) At 24 hours post-treatment, cells were harvested to assess ISG induction. qRT-PCR analysis of ISGs (Mx1, IFIT1, and ISG15) was performed following treatment by conditioned media. Relative expression was normalized to TBP. Data represent n ≥ 3 biological replicates. Statistical significance was determined using two-way ANOVA (*P < 0.05, P < 0.01 **, P < 0.001 ***, P < 0.0001 ****, ns = not significant). Error bars represent standard deviation with the mean as the center.

ISG transcripts levels via qRT-PCR (Fig 7A and 7C). Consistent with our STAT1 activation results (Fig 7B), a robust induction of ISGs, including Mx1, IFIT1, and ISG15, was observed in cells treated with WT-conditioned media (Fig 7C). Conditioned media from IFNλ1 KO cells also elevated ISG expression relative to DMEM-F12, but significantly less than WT-conditioned media (Fig 7C), mirroring the modest decrease in ISGs observed in IFNλ1 KO cells (Fig 5C and 5F). In contrast, conditioned media from IFNλ2/3 KO or IRF3 KO cells failed to induce ISGs, resulting in transcript levels

indistinguishable from media-only controls (Fig 7C). As expected, IFNLR KO cells treated with condition media failed to induce any ISGs (Mx1, IFIT1, and ISG15) in response to any conditioned media tested, including those from WT cells, suggesting that IFNλs, are solely responsible for basal JAK/STAT activation in T84 intestinal epithelial cells (Fig 7B and 7C). To confirm that IFNλs are the dominant antiviral factors, T84 WT-conditioned media was applied to cells depleted of the type I IFN receptor (IFNAR) and induction of ISGs was monitored 24 hours post-treatment. IFNAR depleted cells robustly induced Mx1 upregulation and conferred strong antiviral protection [6] (S9A–S9B Fig).

Together, these findings confirm that IFNλ2/3 are the principal drivers of basal JAK/STAT signaling and ISG expression in intestinal epithelial cells. Moreover, they highlight the essential role of IFNλ2/3 in maintaining mucosal immune readiness through basal interferon signaling, independent of viral infection or other inflammatory stimuli.

## Basal IFNλ2/3, not IFNλ1, is crucial to provide cells basal antiviral protection in a paracrine manner

The loss of ISG expression and the increased viral replication observed in IFNλ2/3 KO cells strongly suggest that basal IFNλ2/3 expression is essential for maintaining an antiviral state by sustaining steady-state ISG levels. To further investigate the functional contribution of basal IFNλ expression to antiviral protection, we aimed to test the antiviral activity of basally produced IFNλ1 and IFNλ2/3. To this end, we performed an additional supernatant transfer assay (Fig 8A). Cell culture supernatants (referred to as conditioned media) of T84 WT, IFNλ1 KO, IFNλ2/3 KO, and IRF3 KO cells were collected and applied to T84 IRF3 KO cells for 24 hours. We used IRF3 KO cells as recipient cells as these cells do not produce IFNs upon viral challenges (S8C–S8D Fig) allowing us to specifically assess the antiviral effects of the basal IFNλs brought with the conditioned media. Following 24-hour pre-treatment with conditioned media, cells were infected with VSV-GFP, VSV-UnaG, and RV-UnaG, and infections were assessed at 7 hours post-infection (hpi) for VSV-GFP and VSV-Luc, and 16 hpi for RV-UnaG (Fig 8A). Pre-treatment with conditioned media from WT and IFNλ1 KO cultures significantly reduced VSV-GFP and VSV-Luc infection compared to cells treated with DMEM-F12 control media (Fig 8B–D). In contrast, conditioned media from IFNλ2/3 KO and IRF3 KO cultures failed to provide this protection, resulting in markedly higher infection levels compared to cells treated with WT media (Fig 8B–D). Likewise, pre-treatment with WT- and IFNλ1 KO-conditioned media significantly limited RV-UnaG infection (Fig 8E and 8F). Importantly, pre-treatment with IFNλ2/3- and IRF3 KO-conditioned media failed to limit RV-UnaG infection as compared to WT-conditioned media and display similar infection levels to DMEM-F12 control media (Fig 8E and 8F). These findings underscore the essential role of basal IFNλ2/3 in protecting the intestinal epithelial cells against virus infection in a paracrine manner. While IFNλ1 contributes modestly to baseline antiviral protection, its absence does not substantially impair host defense.

## Antibody-mediated neutralization revealed a functional hierarchy among basal IFNλ subtypes

To further define the relative contribution of individual IFNλ subtypes to basal antiviral immunity, we employed an antibody-based neutralization strategy using conditioned media. We first validated the specificity of the neutralizing antibodies against IFNλ1, IFNλ2, and IFNλ3 by treating cells with recombinant IFNλ proteins in the presence or absence of specific neutralizing antibodies (S10 Fig). After confirming subtype specificity, conditioned media collected from T84 WT cells was incubated with neutralizing antibodies targeting either IFNλ1, IFNλ2, IFNλ3, or their combinations. The antibody-treated conditioned media was then applied to naïve T84 WT cells to assess JAK–STAT activation (1 h post-treatment) and ISG induction (16 h post-treatment) (Fig 9A). Consistent with our genetic knockout data, depletion of IFNλ2 or IFNλ3 from WT conditioned-media resulted in a marked reduction in STAT1 phosphorylation and a substantial decrease in MX1 expression (Fig 9B and 9C). In parallel, neutralized conditioned media was used to pre-treat T84 IRF3-KO cells prior to VSV-Luc infection to evaluate functional antiviral protection (Fig 9D). Importantly, conditioned media lacking IFNλ2 or IFNλ3 failed to confer antiviral protection, leading to VSV-Luc replication levels comparable to untreated controls (Fig 9E). Notably, combined neutralization of IFNλ2 and IFNλ3 did not further reduce signaling or antiviral activity beyond IFNλ2 or IFNλ3 alone, suggesting their redundant contribution to basal antiviral protection (Fig 9B–9E). In contrast, neutralization of IFNλ1

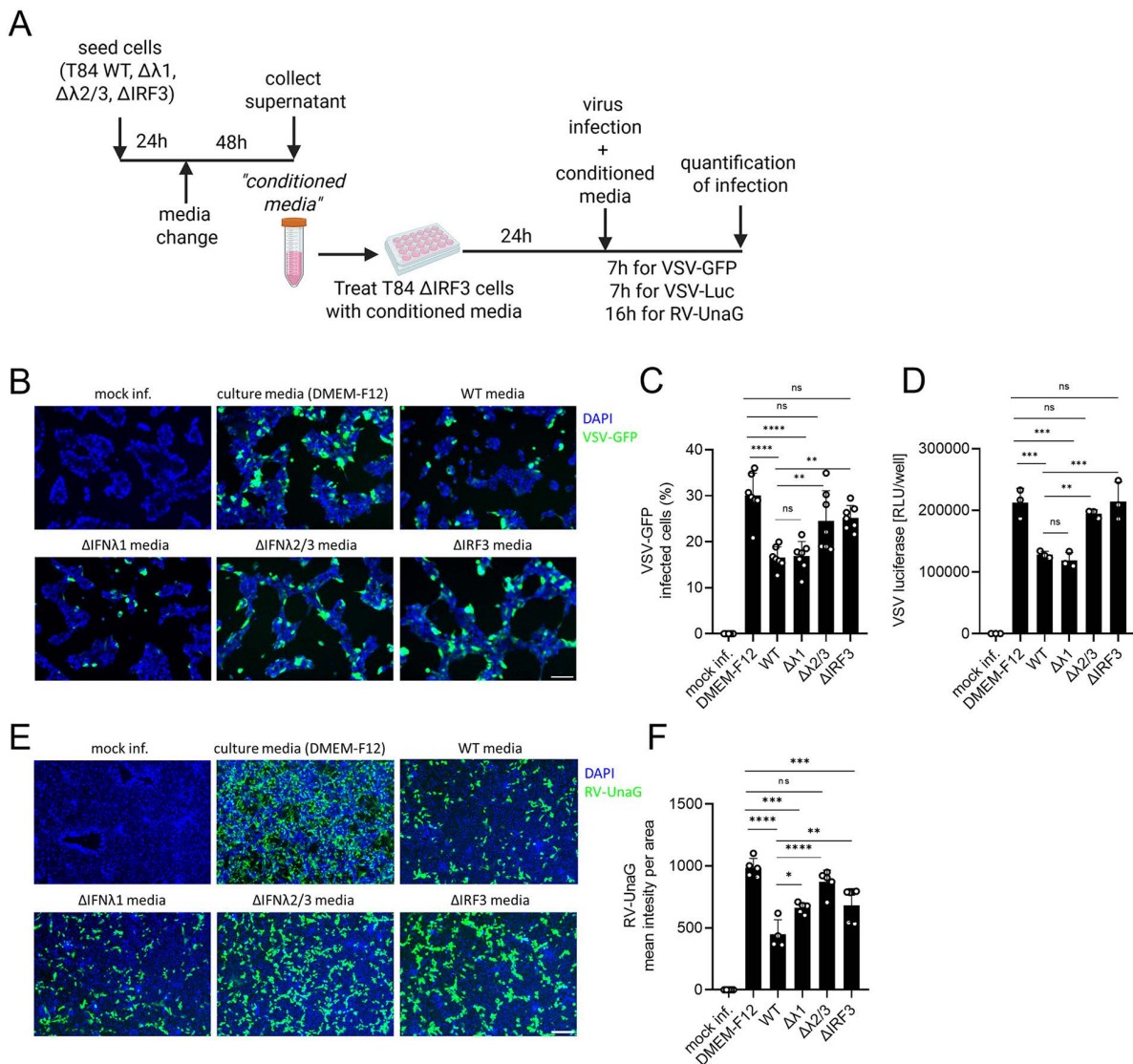

**Fig 8. Basal IFNλ2/3 signaling plays a dominant role in establishing an intrinsic antiviral state.** (A–F) T84 WT, IFNλ1 KO, IFNλ2/3 KO, and IRF3 KO cells were seeded in 6 well plates as 2x10⁶ cells/well, and the media was replaced the following day with 1.5 mL fresh media. Two days later, the cell supernatant was collected after centrifugation at 2000rpm for 5 minutes (referred to as conditioned media), and used to treat T84 IRF3 KO cells for 24 hours. Cells treated with culture media (DMEM-F12) served as a control. At 24 h post-treatment, cells were infected. (A) Schematic representation of experimental design was created in BioRender Keser,Y. (2025) https://BioRender.com/f9bbe51. (B, C) VSV-GFP, (D) VSV_Luc, and (E, F) RV-UnaG. (B) VSV-GFP infection was assessed by live-cell imaging at 7 hpi, with nuclei stained using Hoechst. (C) Quantification of B. (C) VSV-Luc replication was assessed by luciferase assay at 7 hpi. (D) RV-UnaG infection (16 hpi) was evaluated by live-cell imaging, with nuclei stained using Hoechst. (F) Quantification of E. (B, E) Representative images shown. Scale bar = 100 μm. Data represent n ≥ 3 biological replicates. Statistical significance was determined using two-way ANOVA ($P < 0.05$ *, $P < 0.01$ **, $P < 0.001$ ***, $P < 0.0001$ ****, ns = not significant). Error bars represent standard deviation with the mean as the center.

caused only a modest reduction in pSTAT1 and MX1, and IFNλ1-depleted conditioned media largely preserved antiviral activity (Fig 9B–9E). Complete neutralization of all IFNλ subtypes abolished STAT1 activation and ISG induction and fully phenocopied DMEM-F12 controls in antiviral assays, demonstrating that basal interferon-mediated protection in T84 cells is mainly dependent on IFNλ signaling (Fig 9B–9E). Together, these results reveal a clear subtype-specific hierarchy in

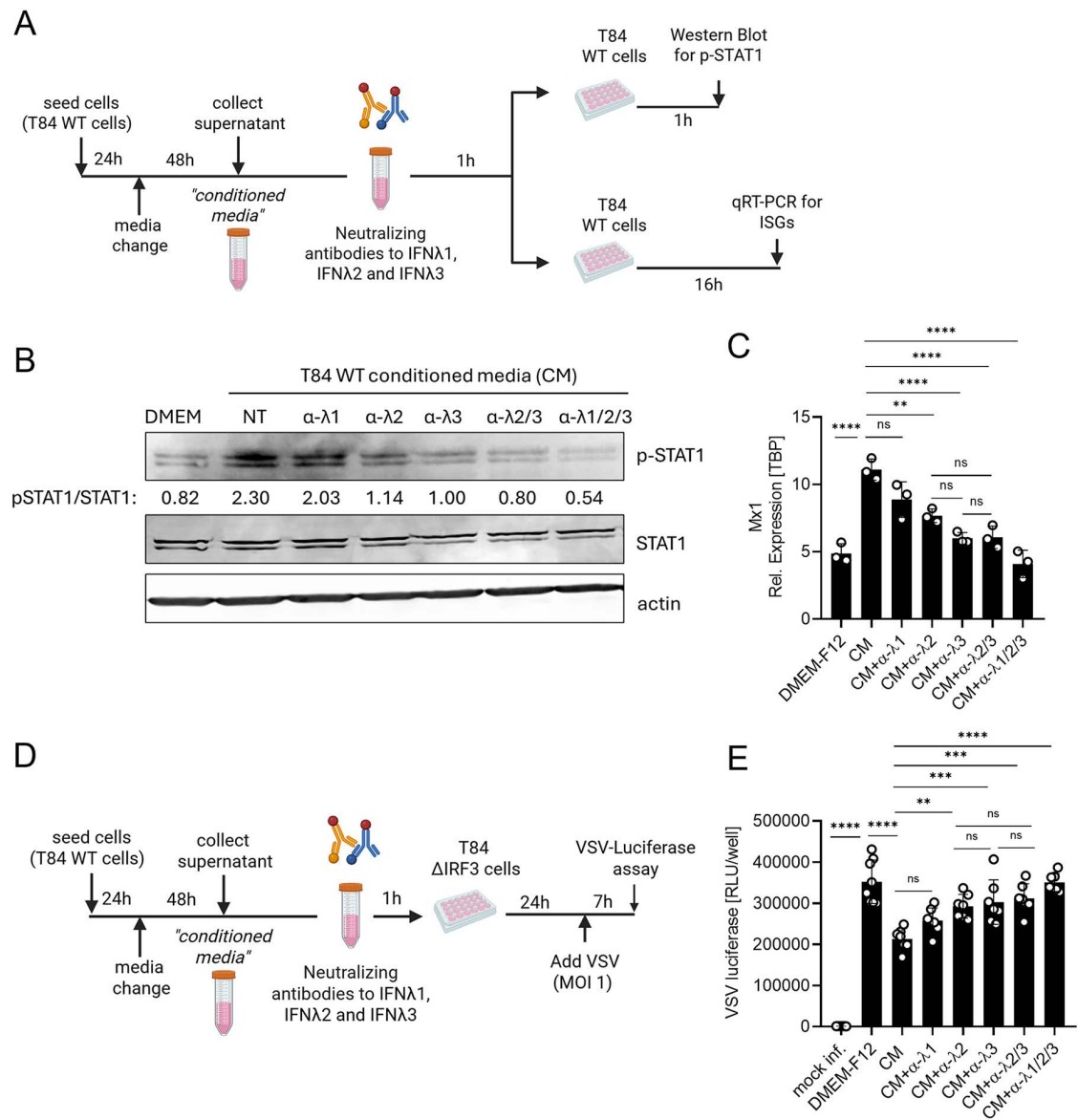

**Fig 9. Neutralization of basal IFNλ3 and IFNλ2, but not IFNλ1, abolishes constitutive JAK–STAT signaling and antiviral protection.** (A) Schematic of the conditioned-media (CM) neutralization workflow was created in BioRender Keser,Y. (2025) https://BioRender.com/drh0ch2. T84 WT cells were seeded in 6 well plates as $2\times10^6$ cells/well, and the media was replaced the following day with 1.5 mL fresh media. Two days later, the cell supernatant was collected after centrifugation at 2000rpm for 5 minutes (referred to as conditioned media (CM)). This conditioned media were incubated with neutralizing antibodies targeting IFNλ1 (α-λ1), IFNλ2 (α-λ2), IFNλ3 (α-λ3), IFNλ2/3 (α-λ2/3), or all three subtypes (α-λ1/2/3) for 1 h at room temperature. Antibody-treated CM were applied to T84 WT cells for analysis of STAT1 phosphorylation (1 h post-treatment) and ISG expression (16 h post-treatment). (B) Representative Western blots showing pSTAT1, total STAT1, and actin as a loading control following treatment with antibody-depleted CM. p-STAT1 protein abundance was quantified relative to STAT1. (C) qRT-PCR analysis of MX1 expression (normalized to TBP) 16 h after antibody-depleted CM treatment. (D) Same as A except CM were used to pre-treat T84 IRF3-KO cells for 24 h prior to VSV-Luc (MOI = 1) infection to assess antiviral activity at 7 hpi. Created in BioRender Keser,Y. (2025) https://BioRender.com/1zuiu9o. (E) VSV-Luciferase assay in T84 IRF3-KO cells pre-treated with antibody-depleted CM at 7 hpi. Data represent n ≥ 3 biological replicates. Statistical significance was determined using one-way ANOVA with multiple-comparison correction (*P < 0.05, P < 0.01 **, P < 0.001 ***, P < 0.0001 ****, ns = not significant). Error bars represent standard deviation with the mean as the center.

basal epithelial immunity: IFNλ3 and IFNλ2 are the principal mediators of constitutive JAK–STAT activation and antiviral defense, whereas IFNλ1 plays only a minor supporting role.

## Replenishment of IFNλ2 and IFNλ3 restores basal antiviral state in IFNλ2/3-deficient cells

To determine whether exogenous IFNλ2 or IFNλ3 can reconstitute basal antiviral signaling in the absence of endogenous IFNλ2/3, we first sought to approximate the physiological amount of each IFNλ subtype present in T84 WT conditioned media. Because IFNλ2/3 protein levels are below the detection limit of ELISA kits, we first used a bioassay-based calibration to identify the range of recombinant IFNλ2 and IFNλ3 concentrations that mimic the antiviral protection conferred by conditioned medium from T84 WT cells (Fig 10A). Titration of recombinant IFNλ2 and IFNλ3 revealed that approximately 1 ng/mL of IFNλ3 closely mimicked the antiviral activity of WT conditioned media, whereas IFNλ2 required slightly higher doses (approximately 5 ng/mL) to achieve comparable protection (Fig 10A and 10B). These findings indicate that IFNλ3 possesses greater antiviral potency than IFNλ2 at equivalent concentrations.

Next, we tested whether chronic supplementation with physiological doses of IFNλ2 (5 ng/mL), IFNλ3 (1 ng/mL), or both could restore basal antiviral signatures in T84 IFNλ2/3 KO cells. After two weeks of continuous IFNλ2 and/or IFNλ3 treatment, cells were seeded in the absence of exogenous IFNλs. Then we assessed their basal ISG expressions by qRT-PCR and Western Blot, and antiviral protection against VSV-Luc (Fig 10C). Chronic IFNλ2 or IFNλ3 supplementation alone efficiently increased levels of IRF7, RIG-I, and STAT1 proteins (Fig 10D), restored ISG expression (MX1, IFIT1, OAS1) (Fig 10E), and substantially reduced VSV-Luc replication in IFNλ2/3 KO cells (Fig 10F). Notably, combined supplementation with IFNλ2 and IFNλ3 did not further increase ISG expression or antiviral protection beyond IFNλ3 alone, indicating that IFNλ3 is the dominant subtype while IFNλ2 provides a secondary, non-additive contribution (Fig 10D–10F).

We previously showed that IFNλ2/3 KO cells respond poorly to IFNλ stimulation compared to T84 WT cells, exhibiting markedly reduced STAT1 phosphorylation, likely due to their intrinsically low total STAT1 abundance (Fig 6C and 6D). To test whether physiological replenishment of IFNλ2/3 can restore this responsiveness, IFNλ2/3 KO cells were chronically supplemented with IFNλ2 (5 ng/mL) and/or IFNλ3 (1 ng/mL) for two weeks. Cells were then reseeded in the absence of exogenous cytokines and acutely re-stimulated with recombinant IFNλ1–3 for 1 hour (Fig 10G). As expected, acute IFNλ1–3 stimulation induced robust STAT1 phosphorylation in T84 WT cells, whereas IFNλ2/3 KO cells showed strongly reduced activation (Fig 10H). Chronic supplementation with IFNλ2 and/or IFNλ3 increased total STAT1 abundance in IFNλ2/3 KO cells, which correspondingly enhanced pSTAT1 levels following acute IFNλ stimulation, indicating a significant restoration of JAK–STAT pathway responsiveness (Fig 10H). Interestingly, downstream ISG expression (MX1, IFIT1, and OAS1) did not follow this pattern. Despite the increase in STAT1 protein and phosphorylation, IFNλ2/3 KO cells chronically supplemented with IFNλ2 and/or IFNλ3 did not exhibit enhanced ISG induction in response to acute IFNλ treatment (Fig 10I). This muted transcriptional response is likely due to persistent negative feedback mechanisms that limit further ISG induction after prolonged exposure to IFNλ. Together, these results demonstrate that both IFNλ2 and IFNλ3 subtypes cooperate to sustain the intact basal antiviral protection. Physiological replenishment of IFNλ2 and/or IFNλ3 is therefore sufficient to reconstruct the basal ISGs and antiviral state lost in IFNλ2/3 KO cells.

## Discussion

Intestinal epithelial cells utilize interferon lambda (IFNλ) as a defense mechanism in response to viral infection [7,53]. However, even in the absence of pathogens, epithelial cells produce basal IFNλs that sustain low-level expression of interferon-stimulated genes (ISGs) in sterile, non-infected conditions [40,54]. Here, we demonstrate distinct and nonredundant roles for basal IFNλ subtypes in establishing an antiviral state, independent of their well-known functions during virus-induced responses. Our work reveals that although both IFNλ1 and IFNλ2/3 are expressed basally, IFNλ2/3 is the dominant driver of the baseline antiviral environment that pre-arms intestinal epithelial cells against viral infection. Transcriptomic profiling showed that IFNλ2/3-deficient cells exhibit globally diminished ISG expression relative to wild-type

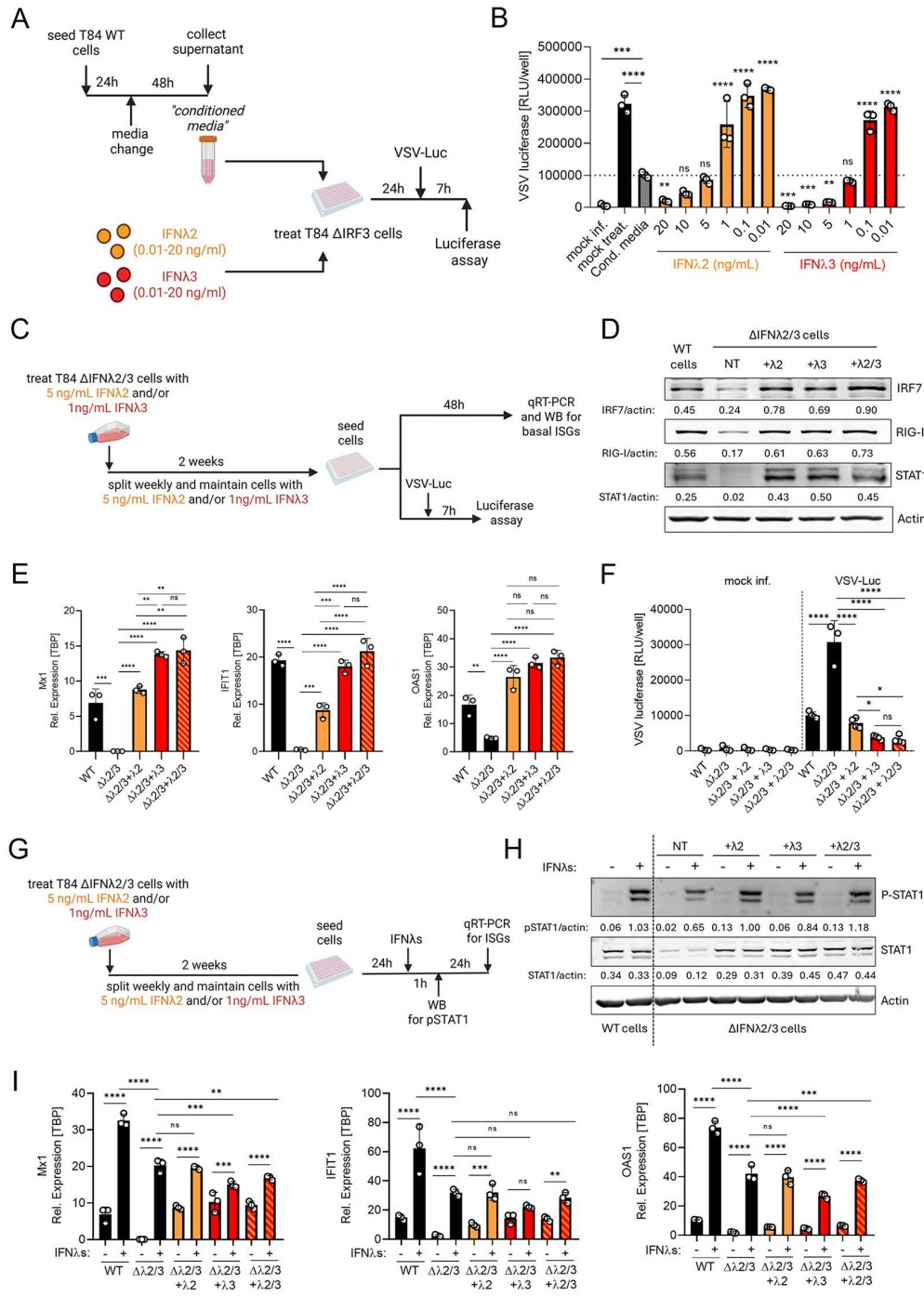

**Fig 10. Chronic IFNλ2 and/or IFNλ3 stimulation in T84 IFNλ2/3 KO cells restores basal ISG expression and antiviral activity.** (A, B) T84 WT cells were seeded, and media was replaced the following day. After 48 h, supernatants (conditioned media) were collected and used as a reference control for antiviral activity. IRF3 KO cells were treated with recombinant IFNλ2 or IFNλ3 (0.01–20 ng/mL) or with WT conditioned media for 24 h and then infected with VSV-Luc for 7 h. (A) Schematic representation of the experimental workflow was created in BioRender Keser,Y. (2025) https://BioRender. com/ip2l074. (B) 7hpi luciferase activity was measured to assess VSV-Luc infection in IRF3 KO cells treated with recombinant IFNλ2 or IFNλ3. (C–F) IFNλ2/3 KO cells were chronically supplemented for two weeks with IFNλ2 (5 ng/mL), IFNλ3 (1 ng/mL), or both. Cells were then trypsinized, reseeded in the absence of any IFN treatment and collected 48 h later for ISG analysis, or used for antiviral assays. (C) Schematic representation of chronic IFNλ2/3 supplementation and subsequent experimental steps. Created in BioRender Keser,Y. (2025) https://BioRender.com/3775duy. (D) Western blot analysis of IRF7, RIG-I, and STAT1 in WT cells and IFNλ2/3 KO cells under the indicated supplementation conditions or non-treated (NT). Protein abundance

was quantified relative to actin. Representative images are shown. (E) qRT-PCR analysis of ISGs (MX1, IFIT1, OAS1) in WT cells and IFNλ2/3 cells maintained with IFNλ2, IFNλ3, IFNλ2 + 3, or non-treated. Relative expression was normalized to TBP. (F) VSV-Luc infection was measured by luciferase assayed 7 hpi in hours in WT and IFNλ2/3 cells maintained with IFNλ2, IFNλ3, IFNλ2 + 3, or non-treated. (G–I) IFNλ2/3 KO cells were chronically supplemented with IFNλ2 (5 ng/mL), IFNλ3 (1 ng/mL), or IFNλ2 + 3 for two weeks, reseeded in the absence of any IFNs, and next day, acutely stimulated with IFNλ1–3 (20 ng/mL of each) for 1 h or 24 h. (G) Schematic representation of chronic supplementation followed by acute IFNλ stimulation, was created in BioRender Keser,Y. (2025) https://BioRender.com/beodbxz. (H) Western blot analysis of p-STAT1 and total STAT1 in WT and IFNλ2/3 cells maintained with IFNλ2, IFNλ3, IFNλ2 + 3, or non-treated (NT). Protein abundance was quantified relative to actin, loading control. Representative images are shown. (I) qRT-PCR analysis of ISGs (MX1, IFIT1, OAS1) 24 h after acute IFNλ1–3 stimulation in WT and ΔIFNλ2/3 cells supplemented as indicated. Relative expression was normalized to TBP. Data represent n ≥ 3 biological replicates. Statistical significance was determined using two-way ANOVA ($P < 0.05$ *, $P < 0.01$ **, $P < 0.001$ ***, $P < 0.0001$ ****, ns = not significant). Error bars represent standard deviation, with the mean shown at the center.

cells, whereas IFNλ1-deficient cells maintain near–wild-type transcriptional signatures (Fig 5). This decreased ISG expression in the absence of basal IFNλ2/3 significantly increased susceptibility of intestinal epithelial cells to multiple viruses derived from diverse families (-ssRNA, dsRNA, and DNA viruses) (Fig 3). Importantly, basal IFNλ2/3 signaling maintains not only ISG expression but also the core machinery required for IFN induction and responsiveness, including IRF7, STAT1, STAT2, and IRF9 (Figs 5F and 6A, B). Consistent with this, IFNλ2/3-deficient cells display reduced total STAT1 levels and impaired responsiveness to exogenous IFNλ (Fig 6C, 6D). Chronic supplementation of IFNλ2/3 knockout cells with low-dose recombinant IFNλ2 or IFNλ3 restored STAT1 expression and IFN responsiveness (Fig 10), underscoring the essential role of basal IFNλ2/3 in priming epithelial cells for robust antiviral signaling. Supernatant-transfer assays further confirmed that constitutively produced IFNλ2/3 drives basal STAT1 phosphorylation and ISG induction through paracrine signaling (Figs 7 and 9). Moreover, conditioned media lacking IFNλ2/3 failed to confer antiviral protection upon recipient cells, in contrast to conditioned media from wild-type or IFNλ1-deficient cultures (Figs 8 and 9). Together, these findings establish that constitutive IFNλ2/3 expression, rather than IFNλ1, is the principal mediator of basal JAK/STAT activation and ISG maintenance in intestinal epithelial cells (Fig 11). Our study provides one of the first detailed demonstrations of basal IFNλ function in human epithelium and offers compelling evidence for distinct physiological roles of IFNλ1 versus IFNλ2/3.

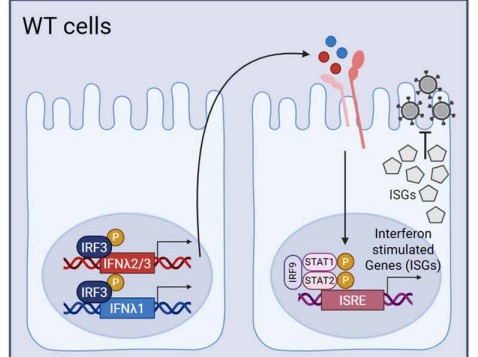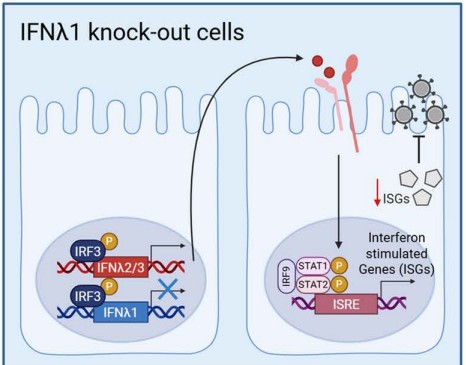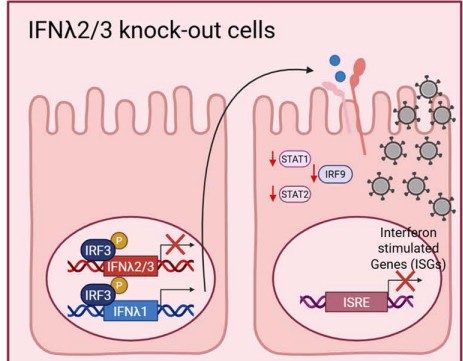

**Fig 11. Basal IFNλ2/3 are essential for maintaining ISG expression and antiviral protection in intestinal epithelial cells.** In WT cells (left panel), both IFNλ1 and IFNλ2/3 are produced under homeostatic conditions via IRF3 activation. Secreted IFNλs engage the IFNLR receptor on neighboring cells, activating the JAK/STAT pathway and inducing robust expression of ISGs thereby limiting viral replication. In IFNλ1 KO cells (middle panel), IFNλ2/3 are still expressed and can activate STAT1/2 signaling and ISG expression, maintaining effective antiviral defense with only a minor reduction in ISG levels. In contrast, IFNλ2/3 KO cells (right panel) retain IFNλ1 expression but exhibit a dramatic loss of STAT1/2 expression and fail to activate ISG transcription, resulting in impaired JAK/STAT signaling and increased viral replication. These findings highlight the predominant and non-redundant role of IFNλ2/3 in establishing and sustaining the basal antiviral state in intestinal epithelial cells. Schematics were created in BioRender Keser,Y. (2025) https://BioRender.com/3oi4mf0.

While we [6,38,39,45,55] and others [51,52,54,56] have detected basal IFNλs in mock-treated samples when evaluating virus infection, their functional significance has not been directly addressed. To date, no studies have thoroughly examined the role of basal IFNλ expression in human epithelial cells or tissues where they are known to play a central role in antiviral defense. Here, we functionally characterize and differentiate basal IFNλ1 and IFNλ2/3 signaling for the first time by demonstrating a marked reduction in key ISG expression in the absence of basal IFNλ2 and IFNλ3, but not IFNλ1. Furthermore, we show that IFNλ2/3 KO cells exhibit diminished responsiveness to paracrine JAK/STAT signaling due to reduced expression of signaling effectors such as STAT1. Previous studies have similarly shown that type I interferons, particularly IFNβ, are constitutively produced at low levels in various cell types via IRF3/7-dependent mechanisms [57,58]. This constitutive IFNβ signaling maintains baseline ISG expression, enhancing antiviral preparedness through continuous expression of critical antiviral proteins (e.g., MX proteins, OAS enzymes, IFIT family members, RIG-I), while also regulating immune cell responsiveness, cellular metabolism, and tissue homeostasis [59,60]. Taken together, these findings underscore the essential role of basal IFNs in antiviral defense and responsiveness. Given the central role of IFNλs at mucosal barriers, dissecting the subtype-specific mechanisms of basal IFNλ signaling is crucial to understanding their contribution to epithelial immune readiness.

Studies characterizing IFNλ function have primarily focused on individual recombinant subtypes (IFNλ1, IFNλ2, IFNλ3 and IFNλ4), but direct comparisons across all IFNλ subtypes remain critically lacking. Our findings reveal that recombinant IFNλ subtypes show comparable antiviral activity against viruses from different families (Fig 2), yet, for the first time, we demonstrate a functional distinction between the antiviral roles of basal IFNλ1 and IFNλ2/3 in intestinal epithelial cells, with IFNλ2/3 playing a dominant, non-redundant role compared to basal IFNλ1 (Fig 8). Previously, we compared WT IFNλ3 with WT IFNλ4 and naturally occurring IFNλ4 variants (P70S and K154E), showing distinct levels of ISG induction and antiviral potency in human hepatocytes and intestinal cells [17]. Furthermore, other studies found differential antiviral activities of IFNλ1 and IFNλ3 against VSV and EMCV in hepatocytes, with IFNλ3 potently restricting EMCV while IFNλ1 showed weak control, and conversely, IFNλ1 strongly inhibiting VSV whereas IFNλ3 had no effect [11]. These findings underscore that type III IFNs are not functionally equivalent, although the underlying mechanisms remain unclear. Notably, receptor binding studies have shown IFNλ3 has the highest affinity for the receptor complex, while IFNλ2 has the lowest [11,61,62], yet it remains unresolved whether receptor affinity alone accounts for differences in antiviral activity. A distinguishing feature of our study is the use of both recombinant IFNλ treatment (paracrine signaling) and genetic knockout approaches to dissect subtype-specific roles of IFNλs. A key insight from our work is the discrepancy between the similar antiviral effects of recombinant IFNs and the divergent outcomes seen upon endogenous IFNλ depletion. Treatment with recombinant IFNs introduces each IFNλ subtype in a controlled and acute manner, often at non-physiological levels, thereby inducing a robust antiviral state. In contrast, genetic depletion of IFNλ subtypes uniquely enabled evaluation of cell-intrinsic IFNλ responses, including physiological consequences of receptor binding, thereby capturing differences in their endogenous functions. These findings highlight a critical limitation of relying solely on recombinant antiviral assays, which may not capture the complete antiviral functions of individual IFNλ subtypes.

Our data suggest that basal IFNλ1 contributes minimally to the maintenance of basal antiviral immunity in intestinal epithelial cells, while IFNλ2/3 plays a dominant role (Figs 4–9). However, previous studies have shown that IFNλ1 and IFNλ3 are both crucial and exhibit virus-specific antiviral activities in hepatocytes [11]. This indicates that IFNλs may play more nuanced, tissue-specific functions that depend on both the site of infection and the dynamics of infection. Moreover, it raises the possibility that basal and virus-induced IFNλ responses may serve distinct functions. The functional distinction of IFNλ1, IFNλ2, and IFNλ3 may stem from their evolutionary history. Unlike IFNλ2 and IFNλ3, which are conserved across a broad range of mammals, IFNλ1 shows more limited conservation and is functional primarily in primates. In several non-primate species, including mice, IFNλ1 is pseudogene and appears to have emerged later in evolution, likely through gene duplication and divergence events specific to primates [13,14]. This restricted phylogenetic distribution suggests that IFNλ1 may have evolved to serve more specialized roles in primates, potentially tailored to unique

host-pathogen interactions. Collectively, these findings support the idea that IFNλ subtypes are not functionally equivalent and that their differences are highly context dependent. Further investigation is needed to elucidate the distinct antiviral mechanisms of each subtype. A deeper understanding of IFNλ biology, particularly the distinct roles of basal versus virus-induced signaling, is essential for fully capturing the complexity of type III IFN antiviral activities and for guiding the therapeutic use of IFNλ.

In conclusion, our study establishes that constitutive IFNλ2/3 expression, rather than IFNλ1, is the primary mediator of baseline antiviral immunity in the intestinal epithelial cells. This is achieved through the maintenance of basal JAK/STAT pathway activation, robust ISG expression, and the priming of cells for effective responses to viral challenges and to paracrine IFN signaling. These findings highlight a clear functional hierarchy among IFNλ subtypes, positioning IFNλ2/3 as essential regulators of mucosal antiviral immunity and offering a foundational framework for developing targeted therapeutic strategies against enteric viral infections. Due to technical limitations, the phenotypes obtained through our genetic knockout approaches could not be verified in primary intestinal epithelial models or organoid cultures. We therefore acknowledge this as a limitation, and future studies will be needed to determine whether these basal IFNλ phenotypes are preserved in primary systems. Future research should focus on understanding the precise mechanisms that lead to the differential basal activity of IFNλ1 versus IFNλ2/3 including investigating the specific transcriptional or post-transcriptional regulatory elements that govern their distinct contributions will be critical to understand their functions.

## Methods

### Cell lines and cell culture

Wild type (WT) T84 (ATCC CCL-248) as well as T84 knock-out (KO) cells were cultured in a 50:50 mixture of Dulbecco's Modified Eagle's Medium (DMEM) and F12 (Gibco #11320033). Lenti-X HEK293T cells (Takara, 632180) were grown in DMEM (Gibco #31965). All media was supplemented with 10% fetal bovine serum (FBS) (Sigma Aldrich #12306 C) and 100 U/mL penicillin and 100 µg/mL streptomycin (Gibco #15140122). All cell lines were authenticated by STR profiling. All cell lines were tested for mycoplasma contamination biweekly by PCR on culture supernatants using DreamTaq DNA polymerase (5 U/µL, Cat# EP0701, Thermo Scientific) with the MW28 (5′-CCAGACTCCTACGGGAGGCA-3′) and MW29 (5′-TGCGAGCATACTACTCAGGC-3′) primers (50 µM each), which amplify a ~500 bp product. PCR products were resolved on 2% agarose in 0.5×TBE and visualized by ethidium bromide staining under UV light. T84 WT and KO cells were cultured on collagen coated surfaces. Plastic surfaces (*e.g.,* culturing flasks and multi-well plates) were coated with 0.01 mg/mL rat tail collagen (Sigma Aldrich #C7661) diluted in 60% EtOH for 1 hour at 37°C. Collagen was removed and surfaces were washed 2X in PBS prior to seeding cells. All cells were kept in a constant humid atmosphere at 37°C, 5% $CO_2$, and 21% oxygen. For splitting, 0.25% Trypsin-EDTA (Gibco #25200056) was used for T84 cells which were split in a 1:2 ratio, and 0.05% Trypsin-EDTA (Gibco #25200054) was used for HEK293T which were split in a 1:10 ratio.

For all experiments, unless otherwise stated, cells were maintained at approximately 70–80% confluence. For chronic supplementation with IFNλ2 and/or IFNλ3, IFNλ2/3 KO cells were seeded in the presence of 5 ng/mL IFNλ2 (R&D Systems #1587IL025/CF) and 1 ng/mL IFNλ3 (R&D Systems #5259-IL-025/CF). The cytokine-containing media were refreshed every two days. Once per week, cells were passaged at a 1:3 ratio.

### Generation of T84 knock-out cell lines

T84 IFNLR KO [6], and IRF3 KO [58] cell lines were previously generated in our laboratory using CRISPR-Cas9 gene editing approach. The T84 IFNλ1 KO and IFNλ2/3 KO cell lines were generated using a lentivirus-based CRISPR-Cas9 gene editing approach. Briefly, single-guide RNAs (sgRNAs) targeting the coding region of IFNλ1, IFNλ2, and IFNλ3 were inserted into the BamHI cloning site of the lentiviral vector lentiCRISPR v2 (Addgene #52961), which originally contains a puromycin resistance gene. To generate vectors with blasticidin resistance for IFNλ2 and IFNλ3 targeting, the

puromycin resistance cassette was replaced with a blasticidin resistance gene using Gibson Assembly. The following sgRNAs were used: fw: 5'-CACC**GGGAACTCACCAAGGCGTCCC**-3', rev: 5'-AAAC**GGGACGCCTTGGTGAGTTCCC**-3' for IFNλ1, fw: 5'-CACC**GTGGGGACTGCACGCCAGTGC**-3', rev: 5'-AAAC**GCACTGGCGTGCAGTCCCCAC**-3' for IFNλ2, and fw: 5'-CACC**GCTGGAGCAGTTCCTGTCGCC**-3', rev: 5′-AAAC**GGCGACAGGAACTGCTCCAGC**-3' for IFNλ3 (gene targeting sequence in **bold**). To generate the lentiviruses, Lenti-X HEK293T cells seeded at 80% confluency in a 10 cm$^2$ dish were transfected using the transfection reagent Polyethylenimine (PEI) (Polysciences #23966–100) at a PEI:DNA ratio of 4:1. 8 µg of the lentiCRISPR v2 containing the sgRNA targeting IFNλ1, IFNλ2, or IFNλ3, 4 µg pMDG.2 plasmid (Addgene #12259), and 4 µg psPAX (Addgene #12260) plasmid were used for the transfection of each 10 cm$^2$ dish. Three days post transfection, the supernatant was collected, spun down at 4000 rcf for 10 min, and filtered through a 0.45 µm syringe filter (Lab Unlimited #W10462100). To pellet the lentiviruses, the supernatant was spun down at 125,000 rcf for 1.5 h using a SW40 Ti rotor. The lentivirus pellet was resuspended in 100 µL of OptiMem (Gibco #31985062) (per 10 cm$^2$ dish of HEK293T cells). For the lentiviral transduction, 3x10$^5$ T84 WT cells were seeded per well of a 6-well plate. 16 hours post-seeding, cells were transduced with 20 µL of the concentrated lentiviruses supplemented with 3 µL Polybrene infection/transfection reagent (Sigma Aldrich #TR-1003-G) diluted in 3 mL of DMEM-F12 media. Three days post-transduction, transduced cells were selected with 0.1 mg/mL Puromycin (Invitrogen #ant-pr-1) or 0.1 mg/mL Blasticidin (Invitrogen #ant-bl-1). Single cell cloning was performed using a limited serial dilution approach in 96-well plate. KO of IFNλ1, IFNλ2, and IFNλ3 genes was confirmed by Sanger sequencing (see details below) (S1A Fig) and functional assays (S1B, S1C Fig).

## Genomic DNA isolation, PCR, and gel extraction

T84 WT, IFNλ1 KO, and IFNλ2/3 KO cells were harvested, and genomic DNA (gDNA) was isolated by using Monarch Genomic DNA Purification Kits (NEB #T3010S) according to manufacturers' protocol. To amplify the genomic loci of IFNλ1, IFNλ2, and IFNλ3, PCR amplification was performed using 5X Phusion HF Buffer (Thermo Fisher #F-549S) and Phusion Hot Start II DNA Poly (Thermo Fisher #F518L) with the following primers: IFNλ1 fw: 5'-GTTGCGATTTAGCCATGGCTGCAGCTTGGAC-3', IFNλ1 rev: 5'-AACTCAGCCCTATGTCTCAGTCAGGGCTGCA-3', or IFNλ2 and IFNλ3 fw: 5'-CTAGGTGAGTCCCACATCTCTGTCCGTGCTCAG-3', IFNλ2 rev: 5'-CCTGGAGGTGAGTTGGATTTACACACAC-3' (same forward primer used for both), IFNλ3 rev: 5'-GCGACTGGGTGACAATAAATTAAGCCAAGTGGC-3'. The PCR products were subjected to electrophoresis on 1% Agarose (Sigma-Aldrich #A6013) in 1X TBE (1.1M Tris-base, 900mM Boric Acid, 25mM EDTA). Specific amplicon bands (IFNλ1: 2215 bp, IFNλ2: 1555 bp, IFNλ3: 1516 bp) were extracted using the Monarch DNA Gel Extraction Kit (NEB #T1020S). Extracted DNAs were sequenced using Sanger sequencing (GENEWIZ, Azenta Life Sciences).

## Viruses and viral infection

SA11 rotavirus encoding the green fluorescent protein UnaG fused to the NSP3 gene (RV-UnaG) was a kind gift from John Patton, Indian University, and was amplified and semi-purified as previously described [63]. Mammalian reovirus (MRV) type 3 clone 9, originally obtained from Bernard N. Fields, was propagated and purified following standard protocols [64]. Vaccinia virus expressing eGFP (VV-GFP), a Western Reserve strain with eGFP under the control of a synthetic early/late promoter, was first described by Mercer and Helenius [65] and was kindly provided by Jason Mercer. VV-GFP was grown and purified using standard methods [66]. Vesicular stomatitis virus expressing GFP (VSV-GFP) and luciferase (VSV-Luc) were generous gifts from Sean Whelan (Washington University) and was propagated as previously described [67].

For infection assays, all virus infections were performed at the multiplicities of infection (MOIs) indicated in figure legends. MRV, RV-UnaG, VSV-GFP, and VV-GFP infections were conducted by diluting virus stocks in complete culture

medium and incubating cells for the indicated durations. Prior to infection, rotavirus was activated at 37°C for 30 min in serum-free medium containing 2 µg/mL trypsin from bovine pancreas (Sigma #T1426). Cells were washed twice with serum-free medium before virus inoculation and subsequently incubated for 1 h at 37°C to facilitate infection.

## VSV-luciferase assay

T84 cells were seeded in 96-well plates at a density of 30,000 cells/well and allowed to adhere overnight. The following day, cells were infected with VSV expressing luciferase (VSV-Luc, MOI = 1) in serum-free media. At 7 hours post-infection (hpi), the media was removed, and cells were lysed using 100 µL of Passive Lysis Buffer (Promega #E1941). Luciferase activity was measured using the Bright-Glo Luciferase Assay System (Promega #E2610) according to the manufacturer's instructions. Briefly, 95 µL of cell lysate was transferred to a white-walled 96-well plate, followed by the addition of 95 µL of Luciferase Assay Reagent II. Luminescence was measured immediately using a luminometer (Tecan #33804).

## Poly I:C transfection

T84 cells were transfected with polyinosinic:polycytidylic acid (poly I:C) using Lipofectamine 2000 (Invitrogen #11668019). Cells were seeded in 48-well plates at a density of $3 \times 10^5$ cells per well and allowed to reach ~80% confluency before transfection. For transfection, poly I:C (InvivoGen #tlrl-pic) was diluted in Opti-MEM (Gibco #31985070) to a final concentration of 1 µg/mL. Lipofectamine 2000 was separately diluted in Opti-MEM at a 1:25 dilution and incubated for 5 minutes at room temperature. The diluted poly I:C was then mixed with the diluted Lipofectamine 2000 at a 1:1 ratio and incubated for 20 minutes at room temperature to allow complex formation. The transfection mixture was added dropwise to cells and incubated for 6 hours at 37°C in 5% $CO_2$.

## Neutralization of IFN-λ1, IFN-λ2, and IFN-λ3 in conditioned media

Basal levels of IFNλ1, IFNλ2, and IFNλ3 present in conditioned media were selectively depleted using the capture and detection antibodies provided in the corresponding DuoSet ELISA kits (R&D Systems, Bio-Techne # DY7246, #DY1587, # DY5259). For each interferon, neutralization was performed individually. Capture and detection antibodies were reconstituted according to the manufacturer's instructions and diluted 1:100 in conditioned media, and incubated at room temperatures for 1 hour, then immediately applied to recipient cells.

## RNA isolation, cDNA synthesis, and qRT-PCR

Total RNA was extracted using the RNeasy Mini Kit (Qiagen #74136) following the manufacturer's protocol. RNA concentration and purity were assessed using a NanoDrop spectrophotometer (Thermo Scientific). For cDNA synthesis, 250 ng of total RNA was reverse-transcribed in a 20 µL reaction using 4 µL 5x iScript Reaction Mix and 1 µL iScript Reverse Transcriptase (Bio-Rad #1708890) for RT-qPCR. The reverse transcription program was carried out as follows: 5 min at 25 °C (priming), 20 min at 46 °C (reverse transcription), 1 min at 95 °C (enzyme inactivation). Quantitative RT-PCR (qRT-PCR) was performed using iTaq Universal SYBR Green Supermix (Bio-Rad #1725124) in a 15 µL reaction volume containing: 7.5 µL SYBR Green Supermix, 3.8 µL each of forward and reverse primers diluted 1:100 (final concentration: 250 nM), 2 µL of cDNA diluted 1:2, 1.7 µL nuclease-free water.

 qPCR reactions were carried out on the CFX Opus 96 Real-Time PCR System (Bio-Rad #12011319) with the following cycling conditions: 95 °C for 30 sec (initial denaturation), 40 cycles of: 95 °C for 5 sec (denaturation), 60 °C for 30 sec (annealing/extension), followed by a melt curve analysis to confirm specificity of amplification. The expression of target gene was normalized to the housekeeping gene TaTa box binding protein (TBP). Primer sequences are listed in Table 1.

**Table 1. List of primer sequences used in qPCR.**

| Target Gene | Forward sequence (5'→3') | Reverse sequence (5'→3') |
|---|---|---|
| Human IFIT1 | AAAAGCCCACATTTGAGGTG | GAAATTCCTGAAACCGACCA |
| Human IFNλ2/3 | GCCACATAGCCCAGTTCAAG | TGGGAGAGGATATGGTGCAG |
| Human IFNλ1 | GCAGGTTCAAATCTCTGTCACC | AGCTCAGCCTCCAAGGCCACA |
| Human IRF7 | TCTTCTTCCAAGAGCTGG | CTATCCAGGGAAGACACAC |
| Human ISG15 | CCTCTGAGCATCCTGGT | AGGCCGTACTCCCCCAG |
| Human Mx1 | GAGCTGTTCTCCTGCACCTC | CTCCCACTCCCTGAAATCTG |
| Human OAS1 | TGCGCTCAGCTTCGTACTGA | GGTGGAGAACTCGCCCTCTT |
| Human RIG-I | TTGCAATATCCTCCACCACA | GGCATGTTACACAGCTGACG |
| Human TBP | CCACTCACAGACTCTCACAAC | CTGCGGTACAATCCCAGAACT |

## SDS-PAGE and western blot

Adherent cells were lysed in 1X RIPA buffer [150 mM NaCl, 50 mM Tris-HCl (pH 7.4), 1.0% Triton X-100, 0.5% sodium deoxycholate, and 0.1% SDS] supplemented with cOmplete Mini EDTA-free Protease Inhibitor Cocktail and phosphatase inhibitor PhosSTOP for 5 min at 37 °C. Lysates were collected and protein concentration was measured using the Pierce BCA Protein Assay Kit assay (Thermo Scientific #23225) according to the manufacturer's instructions. 5–10 µg protein per condition were separated by SDS-PAGE and blotted onto a 0.2 µm nitrocellulose membrane (Bio-Rad #1704158) using a Trans-Blot Turbo Transfer System (Bio-Rad). Membranes were blocked with Tris Buffer saline (TBS)-tween (0.5% Tween in TBS) containing 5% Bovine Serum Albumin (BSA) or containing 50% Intercept (TBS) Blocking Buffer (Licor #927–60001) for 1–2 h at room temperature (RT). Membranes were incubated with primary antibodies against IRF3 (Cell Signaling #11904T), Mx1 (Santa Cruz #sc-271024), IRF7 (Cell Signaling Technologies # 5184S), RIG-I (AdipoGen # AG-20B-0009), ISG15 (Santa Cruz #166755), STAT1 (BD Biosciences #610115), phospho-STAT1 (BD Biosciences #612233), and actin (Sigma Aldrich #A5441) diluted 1:1000 in blocking buffer overnight at 4°C. Anti-mouse-IgG (Abcam #ab6789) and anti-rabbit-IgG (Abcam #ab97051) antibodies coupled with horseradish peroxidase (HRP) (GE Healthcare #NA934V) or IRDye 680RD/800CW (Licor 926–68073/ 926–32210) were used as secondary antibodies. Membranes were washed three times with TBS-T for 5 min at RT after each step. The Pierce ECL Western Blotting Substrate (Thermo Fisher #32209) was used for detection of HRP conjugated antibodies according to manufacturer instructions. The membranes were imaged with the ImageQuant LAS 4000 (GE Healthcare) or Odyssey M imaging system (Licor). Quantification was done using the open image analysis software ImageJ. Relative abundance of target protein was normalized to the loading control housekeeping protein, actin.

## Enzyme-linked immunosorbent assay (ELISA)

The levels of IFNλ1 and IFNλ2/3 in cell culture supernatants were measured using the Human IL-29/IFN-lambda 1 DuoSet ELISA (biotechne rd systems # DY7246) and DIY Human IFN-Lambda 2/3 (IL-28A/B) ELISA (pbl assay science #61830) kit, following the manufacturer's protocol. Briefly, half-area high-binding 96-well plates (Fisher #07000091) were coated with the capture antibody overnight at 4°C. After blocking with assay buffer, samples and standards were added in duplicate and incubated at room temperature. Following extensive washing, the detection antibody was applied, followed by incubation with streptavidin-HRP. Signal development was carried out using the substrate solution (BD Biosciences # 555214), and absorbance was measured at 450 nm using a microplate reader (BioTek #BT800TS). IFNλ1 and IFNλ2/3 concentrations were determined by interpolating sample absorbance values from a standard curve generated using recombinant IFNλ1 and IFNλ2.

## Indirect immunofluorescence staining

MRV infected cells were washed with PBS and fixed in 2% Paraformaldehyde (PFA) (Roth #0335.3) (diluted in PBS) for 20 min at RT. Cells were washed in PBS three times and permeabilized in 0.5% Triton-X100 (Sigma-Aldrich #X100-500ML) diluted in PBS for 15 mins at RT. Cells were blocked using 10% FBS (Sigma Aldrich #12306 C) in PBS for 30 min at RT. Primary antibody against MRV µNS [55] diluted in 10% FBS (in PBS) and incubated for 1 h at RT. Cells were incubated with Alexa Fluor 488 conjugated secondary antibody with DAPI (Invitrogen #P36941), both diluted in 1% BSA in PBS for 1 h at RT. Cells were washed in PBS three times after each step. Samples were imaged on a ZEISS Celldiscoverer 7.

## Fluorescence imaging and image analysis

Live-cell imaging of cells was acquired using the epifluorescent ZEISS Celldiscoverer-7 (CD7) Widefield microscope. To assess viral infection in real time, live-cell microscopy was performed. T84 WT, IFNλ1 KO, and IFNλ2/3 KO cells were seeded in 48-well plates and infected with RV-UnaG, VSV-GFP, or VV-GFP. At the experimental endpoint, cells were washed with culture medium and incubated with Hoechst nuclear dye for 30 minutes at 37°C and 5% $CO_2$. Cells were then imaged immediately using live-cell microscopy. To monitor viral infection dynamics in real-time, live-cell fluorescence microscopy was performed. T84 WT, IFNλ1 KO, and IFNλ2/3 KO cells were seeded in 48-well plates and maintained at 37°C with 5% $CO_2$ throughout the entire imaging procedure. Cells were infected with RV-UnaG, VSV-GFP, or VV-GFP and imaged every 30 minutes over a 48-hour period using live-cell microscopy. Images were acquired using a 20× objective with 0.5× optical zoom. The laser was set to 50% intensity, and images were captured with a 200 ms exposure time for AF488 channel (for virus). Mean fluorescence intensity (MFI) of GFP and UnaG was quantified using ImageJ Fiji, and MFI values were plotted against time to assess viral replication kinetics.

## Cytotoxicity assay

Cytotoxicity was evaluated using the LDH-Glo Cytotoxicity Assay (Promega #J2380) according to the manufacturer's instructions. T84 WT and KO cells were seeded in 96-well plates and cytotoxicity assays were performed two days later. As a positive control for cytotoxicity, T84 WT cell treated with PPMP (50 µM; D-threo-1-Phenyl-2-hexadecanoylamino-3-morpholino-1-propanol hydrochloride) (Cayman Chemicals #17236) was included. At the indicated time points, 50 µL of culture supernatant was transferred to a clear, flat-bottom 96-well plate to measure extracellular (released) LDH. To quantify intracellular LDH, 15 µL of 10× Lysis Buffer was added to the remaining cells in the original wells, followed by a 1-hour incubation at 37°C. Subsequently, 50 µL of the resulting lysate was transferred to a separate 96-well plate. For both supernatant and lysate samples, 50 µL of LDH Substrate Mix was added, mixed gently, and incubated for 30 minutes at room temperature in the dark. The reaction was terminated by adding 50 µL of Stop Solution to each well. Luminescence was recorded at 490 nm using an 800TS Microplate Reader (BioTek). Cytotoxicity was calculated by normalizing the amount of released LDH to the total intracellular LDH.

## Bulk RNA-sequencing

T84 WT, IFNλ1 KO, IFNλ2/3 KO, and IFNLR KO cells were seeded in a 48-well plate. For each cell line, three biological replicates were prepared. RNA extraction was performed using the Qiagen RNeasy Plus Mini Kit following the manufacturer's instruction. RNA sequencing was performed by GENEWIZ (Azenta Life Sciences). Briefly, cDNA libraries were prepared using the TruSeq Stranded mRNA Library Prep Kit (Illumina) and sequenced on an Illumina NovaSeq 6000 platform, generating 150 bp paired-end reads. Data preprocessing together with quality control report (FAST QC Report) was provided by Genewiz. The quality control of raw sequencing reads was checked using FastQC to ensure high-quality sequencing data. Low-quality reads and adapter sequences were trimmed using Trimmomatic. The cleaned reads were

then aligned to the human reference genome (GRCh38) using STAR aligner. Gene expression levels were quantified using featureCounts, and differential gene expression analysis was conducted with DESeq2. Genes with an adjusted p-value < 0.05 and a fold change ≥ 2 were considered significantly differentially expressed. Gene Ontology (GO) enrichment analysis was performed on the differentially expressed genes using the clusterProfiler package in R.

## Data plotting and statistics

If not specified otherwise, data plotting and all statistical analyses were performed with the GraphPad Prism 5.0 software. The number of biological replicates and statistical tests used are specified in the figure legends. Statistical tests are listed in figure legends. If not specified otherwise, all schematics and illustrations were created with BioRender.com. All figures were assembled with the Affinity Designer 1.10.0 software.

## Supporting information

**S1 Fig. Validation of T84 IFNλ1 KO and IFNλ2/3 KO cells.** (A) Genomic DNA (gDNA) from T84 WT, IFNλ1 KO, and IFNλ2/3 KO cells were isolated, and PCR was performed for the amplification of IFNλ1, IFNλ2, and IFNλ3 loci. PCR products were sequenced by Sanger sequencing. The Basic Local Alignment Search Tool (BLAST) was used to compare the sequences. (B,C). T84 WT, IFNλ1 KO, and IFNλ2/3 KO cells were seeded in 48-well plates and transfected with poly I:C for 6 hours. Cell supernatants were collected and analyzed by ELISA to measure (B) IFNλ1 and (C) IFNλ2/3 protein levels. Data represent n ≥ 3 biological replicates. Statistical significance was determined using two-way ANOVA ($P < 0.0001$ ****, ns = not significant). Error bars represent standard deviation with the mean as the center.
(TIF)

**S2 Fig. Genetic depletion of IFNλ2/3, but not IFNλ1, increases the spread of diverse virus types in intestinal epithelial cells.** T84 WT, IFNλ1 KO, and IFNλ2/3 KO cells were seeded in 96-well plates and infected two days later with (A) VSV-GFP, (B) RV-UnaG, and (C) VV-GFP. Viral spread was monitored using live-cell microscopy every 2 hours for 48 hours. (left) Representative brightfield (gray) and fluorescence images show infection (green) at 0, 12, 24, 36, and 48 hpi. Representative images shown. Scale bar = 250 µm. (right) Mean fluorescence intensity over time per field of view was quantified using ImageJ Fiji. Data represent n ≥ 3 biological replicates. Statistical significance was determined using two-way ANOVA ($P < 0.0001$ ****, ns = not significant). Error bars represent standard deviation with the mean as the center.
(TIF)

**S3 Fig. T84 polyclonal IFNλ2/3 KO cells, but not IFNλ1 polyclonal cells, showed enhanced infection by VSV-GFP and RV-UnaG.** (A-C) T84 WT, IFNλ1 KO polyclonal (pc.), and IFNλ2/3 KO polyclonal cells (pc.) were seeded in 48-well plates, and next day, transfected with poly I:C for 6 hours. Cell supernatants were collected and analyzed by ELISA to measure (A) IFNλ1; (B) IFNλ2; (C) IFNλ3 protein levels. (D-G) T84 WT, IFNλ1 KO polyclonal (pc.), and IFNλ2/3 KO polyclonal (pc.) cells were seeded in 48-well plates. The following day, cells were infected with (D, E) VSV-GFP (MOI = 1) for 7 hours and (F, G) RV-UnaG (MOI = 1) for 16 hours. Nuclei were stained with Hoechst (blue), and infected cells are shown in green. (D, F) Representative images and (E, G) corresponding quantification (right) are shown for each virus. Scale bar = 100 µm. Data represent ≥3 independent biological replicates. Statistical significance was determined by one-way ANOVA (*$P < 0.05$, ***$P < 0.001$, ****$P < 0.0001$, ns = not significant). Error bars represent standard deviation with the mean as the center.
(TIF)

**S4 Fig. Type III IFN signaling plays a more dominant role in controlling virus infection compared to type I IFN signaling in T84 cells.** T84 WT, IFNAR KO, and IFNLR KO cells were seeded in 48-well plates and infected the following day. (A-D) Cells were infected with MOI (=1) (A) VSV-GFP for 7 hours, (B) RV-UnaG for 16 hours and (C) VV-GFP for 16

hours by live-cell microscopy. Nuclei were stained with Hoechst (blue), and infected cells are shown in green. (A) Representative images and (B-D) corresponding quantification are shown for each virus. Scale bar = 100 µm. Data represent ≥3 independent biological replicates. Statistical significance was determined by two-way ANOVA (*$P < 0.05$, ***$P < 0.001$, ****$P < 0.0001$, ns = not significant). Error bars represent standard deviation with the mean as the center.
(TIF)

**S5 Fig. T84 WT and KO cells do not show differences in apoptosis or cell viability.** (A-C) Based on RNA-sequencing data, Gene Set Enrichment Analysis (GSEA) was performed on selected GO biological process terms associated with IFN response (blue), cytotoxicity (purple), and apoptotic pathways (orange). (A) Comparison of WT vs. IFNLR KO cells. (B) Comparison of WT vs. IFNλ2/3 KO cells. (C) Comparison of WT vs. IFNλ1 KO cells. (D) Cytotoxicity was assessed by measuring LDH release in WT, IFNλ1 KO, IFNλ2/3 KO and IFNLR KO cells. Positive control (pos. ctrl.) indicates cells treated with 50 µM PPMP to induce cytotoxicity. n ≥ 3 biological replicates. Statistical analysis was performed using ordinary one-way ANOVA. ($P < 0.0001$ ****, ns = not significant). Error bars represent standard deviation with the mean as the center.
(TIF)

**S6 Fig. Basal ISG expression is significantly reduced in IFNλ2/3 KO polyclonal cells.** (A–D) qRT-PCR analysis of ISGs (Mx1, IFIT1, Viperin, and ISG15) in T84 WT cells, IFNλ1 KO polyclonal cells (pc.), and IFNλ2/3 KO polyclonal cells (pc.) at two days post-seeding. Relative expression was normalized to TBP. Data represent n ≥ 3 biological replicates. Statistical significance was determined using one-way ANOVA with multiple comparisons ($P < 0.05$ *, ns = not significant). Error bars represent standard deviation with the mean shown at the center.
(TIF)

**S7 Fig. Expression of housekeeping gene TBP and ISGs across cell lines based on RNA-seq data.** Expression levels of selected housekeeping gene (TBP) and ISGs (IFIT1, ISG15, MX1, OAS1) were analyzed across WT and KO cell lines using RNA-sequencing data. Each point represents an individual biological replicate. Expression levels are presented as normalized raw counts.
(TIF)

**S8 Fig. Validation of T84 IRF3 KO cells.** (A, B) T84 WT and IRF3 KO cells were seeded and harvested two days post-seeding to assess IRF3 expression. (A) IRF3 protein levels were analyzed by Western blot, with actin used as a loading control. Representative images shown. (B) Basal IFNλ1 and IFNλ2/3 expression levels were quantified by qRT-PCR in T84 WT and IRF3 KO cells. (C) T84 WT and IRF3 KO cells were seeded and infected with MRV the following day. MRV infection was assessed by immunostaining against the MRV µNS protein at 16 hpi. Representative images show nuclei (blue) and MRV-infected cells (green). Scale bar = 100 µm. (D) Same as C except MRV-induced IFNλ1 and IFNλ2/3 expression was quantified by qRT-PCR in T84 WT and IRF3 KO cells. Gene expression levels were normalized to TBP. Data represent n ≥ 3 biological replicates. Statistical significance was determined using (B) an unpaired t-test between WT and IRF3 KO cells and (D) by two-way ANOVA ($P < 0.01$ **, $P < 0.0001$ ****, ns = not significant). Error bars represent standard deviation with the mean as the center.
(TIF)

**S9 Fig. Basal signaling in T84 cells is mainly mediated by IFNλ signaling, not type I IFN signaling.** (A-B) T84 WT cells were seeded, and the media was changed the following day. Two days later, the cell supernatant was collected (referred to as conditioned media, CM) and used to treat T84 IFNLR KO cells (deficient in IFNλ signaling) and IFNAR KO cells (deficient in type I IFN signaling). Cells treated with culture media (DMEM-F12) served as controls. IFNLR KO cells were additionally pre-treated with recombinant IFNλ1–3 (100 ng/mL each) and IFNAR KO cells were additionally - pre-treated with recombinant IFNβ (2000 IU/mL) for 24 hours prior infection as controls. (A) qRT-PCR analysis of the ISG Mx1

at 24 h post-treatment in IFNLR KO and IFNAR KO cells following CM exposure. Relative expression was normalized to TBP. (B) Following 24 h treatment with CM, cells were infected with VSV-Luc (MOI = 1). At 6 hpi, luciferase assays were performed to assess viral replication. Data represent n ≥ 3 biological replicates. Statistical significance was determined using two-way ANOVA (P < 0.01 **, P < 0.0001 ****, ns = not significant). Error bars represent standard deviation with the mean shown at the center.
(TIF)

**S10 Fig. Specificity of neutralizing antibodies against individual IFNλ subtypes.** Recombinant IFNλ1, IFNλ2, or IFNλ3 (10 ng/mL each) was prepared in 250 μL of DMEM-F12 culture media containing 2.5 μL of the corresponding capture antibody and 2.5 μL of the corresponding detection antibody. Mixtures were incubated for 1 hour at room temperature and then immediately applied to T84 WT cells for 1 hour. Cells were harvested, and Western blot analysis of p-STAT1 was performed. Actin was used as a loading control. Representative images are shown.
(TIF)

## Acknowledgments

We would like to thank the Boulant and Stanifer lab members for the constructive discussions and for proofreading this manuscript.

## Author contributions

**Conceptualization:** Yagmur Keser, Steeve Boulant, Megan L. Stanifer.

**Data curation:** Yagmur Keser, Zehra Sena Bumi, Amelia Perez Valiente, Sorin O. Jacobs.

**Formal analysis:** Yagmur Keser.

**Funding acquisition:** Steeve Boulant, Megan L. Stanifer.

**Investigation:** Yagmur Keser, Zehra Sena Bumi, Amelia Perez Valiente, Sorin O. Jacobs.

**Methodology:** Yagmur Keser.

**Project administration:** Steeve Boulant, Megan L. Stanifer.

**Supervision:** Steeve Boulant, Megan L. Stanifer.

**Visualization:** Megan L. Stanifer.

**Writing – original draft:** Yagmur Keser, Steeve Boulant, Megan L. Stanifer.

**Writing – review & editing:** Yagmur Keser, Steeve Boulant, Megan L. Stanifer.

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
