## [Decision Letter · Decision Letter 0]

5 Oct 2025

Basal IFNλ2/3 signaling is required for ISG expression and viral control in human intestinal epithelial cells

PLOS Pathogens

Dear Dr. Stanifer,

Thank you for submitting your manuscript to PLOS Pathogens. After careful consideration, we feel that it has merit but does not fully meet PLOS Pathogens's publication criteria as it currently stands. Therefore, we invite you to submit a revised version of the manuscript that addresses the points raised during the review process.

Please submit your revised manuscript within 60 days Dec 04 2025 11:59PM. If you will need more time than this to complete your revisions, please reply to this message or contact the journal office at plospathogens@plos.org. Please include the following items when submitting your revised manuscript:

We look forward to receiving your revised manuscript.

Kind regards,

Helen M Lazear, Ph.D.

Academic Editor

PLOS Pathogens

Ashley St. John

Section Editor

Editor-in-Chief

PLOS Pathogens

Editor-in-Chief

PLOS Pathogens

orcid.org/0000-0002-7699-2064

**Additional Editor Comments :**

Overall the reviewers appreciated this study investigating the effects of basal IFN-L signaling and distinguishing the effects of IFNL2/3 vs IFNL1. Reviewers did note some concerns about the KO T84 cells and the specific conclusions that can be drawn from these experiments. Addressing the points raised by reviewers 2 and 3 will require additional experiments to support the study's conclusions.

**Journal Requirements:**

At this stage, the following Authors/Authors require contributions: Yagmur Keser, Sena Bumin, Sorin Jacobs, Amelia Perez Valiente, Steeve Boulant, and Megan Stanifer. Please ensure that the full contributions of each author are acknowledged in the "Add/Edit/Remove Authors" section of our submission form.

https://journals.plos.org/plospathogens/s/submission-guidelines#loc-parts-of-a-submission

4) We do not publish any copyright or trademark symbols that usually accompany proprietary names, eg ©,  ®, or TM  (e.g. next to drug or reagent names). Therefore please remove all instances of trademark/copyright symbols throughout the text, including:

- ® on pages: 19, 21, 22, and 23

- TM on pages: 20, 21, 22, and 23.

5) Please upload all main figures as separate Figure files in .tif or .eps format. For more information about how to convert and format your figure files please see our guidelines:

6) We notice that your supplementary Figures are included in the manuscript file. Please remove them and upload them with the file type 'Supporting Information'. Please ensure that each Supporting Information file has a legend listed in the manuscript after the references list.

7) Some material included in your submission may be copyrighted. According to PLOSu2019s copyright policy, authors who use figures or other material (e.g., graphics, clipart, maps) from another author or copyright holder must demonstrate or obtain permission to publish this material under the Creative Commons Attribution 4.0 International (CC BY 4.0) License used by PLOS journals. Please closely review the details of PLOSu2019s copyright requirements here: PLOS Licenses and Copyright. If you need to request permissions from a copyright holder, you may use PLOS's Copyright Content Permission form.

Potential Copyright Issues:

i) Figures 7a, 8a, and 9. Please confirm whether you drew the images / clip-art within the figure panels by hand. If you did not draw the images, please provide (a) a link to the source of the images or icons and their license / terms of use; or (b) written permission from the copyright holder to publish the images or icons under our CC BY 4.0 license. Alternatively, you may replace the images with open source alternatives. See these open source resources you may use to replace images / clip-art:

Note:  If your figures were made with Biorender, please add this information in the figure legends . Please also confirm that you hold a Premium account and provide a pdf copy of the CC BY 4.0 Licence as provided by BioRender. For instructions on how to generate a CC BY 4.0 license for your figure, please see the guidelines here: https://help.biorender.com/hc/en-gb/articles/21282341238045-Publishing-in-open-access-resources.

If you are using the free assets from BioRender, we are unable to publish these images as they are licenced under a stricter licence than CC BY 4.0. In this case we ask you to remove the BioRender images and replace them with open source alternatives.

See these open source resources you may use to replace images / clip-art:

- https://bioart.niaid.nih.gov/

- https://bioicons.com/

- https://healthicons.org/

- https://scidraw.io/

- https://reactome.org/icon-lib

- https://www.phylopic.org/images

- https://journals.plos.org/plosbiology/article?id=10.1371/journal.pbio.3002395

8) Thank you for stating " "The RNA-seq data generated in this study have been deposited in the NCBI Gene Expression Omnibus (GEO) under accession number GSE296527."  Please note that, though access restrictions are acceptable now, your entire minimal dataset will need to be made freely accessible if your manuscript is accepted for publication. This policy applies to all data except where public deposition would breach compliance with the protocol approved by your research ethics board.

9) Please amend your detailed Financial Disclosure statement. This is published with the article. It must therefore be completed in full sentences and contain the exact wording you wish to be published.

3) If any authors received a salary from any of your funders, please state which authors and which funders.

10) Please revise your current Competing Interest statement to the standard "The authors have declared that no competing interests exist."

**Reviewers' Comments:**

Reviewer's Responses to Questions

**Part I - Summary**

Reviewer #1: This well-written manuscript details studies investigating the relative contributions of IFNL1 versus IFNL2/3 to the recently-identified phenomenon of “basal” (not virus-induced) IFNL signaling in intestinal epithelial cells. Using an IEC cell line with various factors genetically disrupted, multiple viruses, drug treatment, and numerous experimental approaches including transcriptional analysis and supernatant transfers, the authors convincingly demonstrate that IFNL2/3, but not IFNL1, is important for basal ISG expression and initial viral regulation. Overall, experiments are well-conducted with appropriate controls for interpretation, and this is a nice study contributing to a growing field. However, there were a few points/questions that should be addressed.

Reviewer #2: It is well established that IFN lambda is important for antiviral defense of epithelial barriers. A recent publication described basal IFNL production by epithelial cells upon reaching high density, but their effect on antiviral defense was not shown. This study extends the previous report of basal IFNL by determining how it impacts antiviral defense against diverse viruses and which IFNL subtypes may be most impactful in T84 cells. Using genetic knockouts, inhibitors, and supernatant transfer, the authors provide complementary evidence that basal IFNL2/3 (but not IFNL1) can boost antiviral immunity. Overall, the study was well written and clearly presented. However, there were several limitations that could be addressed to strengthen the most novel aspects of the study.

Reviewer #3: In this study, Keser et al dissect the distinct biological functions of IFNλ1 and IFNλ2/3 in conferring antiviral protection in T84 intestinal epithelial cell lines. This question is highly relevant and could uncover specialized mechanisms of mucosal antiviral defense. Using genome-edited cells, they demonstrate viral infection can induce IFNλ1 and IFNλ2/3, and these cytokines have equivalent antiviral activity. However, there seems to be a predominant requirement for basal IFNλ2/3 expression in inducing low-level expression of ISGs that can protect cells against viral infection. While the study provides novel insights, there are a few technical limitations that preclude a firm conclusion that basal ISG expression driven by IFNλ2/3 is the underlying protective mechanism. Further positioning the study in context with prior work from this group and others would also help readers better appreciate how these results fit within the broader framework of IFNλ biology, including the potential influence of culture conditions and cell polarization on basal IFNλ expression and ISG induction.

**Part II – Major Issues: Key Experiments Required for Acceptance**

Reviewer #1: (No Response)

Reviewer #2: Major comments.

1. It is clear that the reported phenotype is not virus specific, but it Is perhaps more important to determine whether the phenotype is specific to T84 cells. Otherwise, this should be indicated as a limitation. T84 cells are a workhorse for mechanistic studies but are far from normal epithelial cells. The amount of basal IFNL seen here seems higher than expected for normal epithelial cells (often not detected in primary epithelial cells or biopsy from healthy intestine).

2. Conclusions made about the relative importance of IFNL2/3 vs IFNL1 is weakened by the fact that IFNL2 and IFNL3 are two genes whereas IFNL1 is a single gene. This is true for assays comparing mRNA abundance as well as comparison of IFNL2/3 vs IFNL1 knockout.

3. Conclusions from figure 4 are not really ruling out a role for virus-triggered responses as concluded in the text. In fact, I would conclude that there is a role for IFNL2/3 triggered by the virus as indicated by the effect of the inhibitor in the IFNL1 knockouts (where virus-triggered IFNL2/3 may be playing a role). Additionally, the JAK inhibitor is going to impact Type I IFN signaling so this experiment may be suggestive at best.

4. There is a convincing phenotype of the IFNL2/3 knockout in diminished response to IFN treatment, but if the authors’ model is correct then this should be rescued by extended culture with low levels of basal IFNL2/3 prior to a subsequent treatment with the dose used in fig. 6D. The supernatant transfer assay is strong, but is not definitive because the effect may be due to combination of factors missing from the supe of IFNL2/3 ko. A definitive experiment would be to add low levels of recombinant IFNL2 or IFNL3 and show that this rescues the effect. This could address another caveat of the genetic studies which is that the knockout of two genes IFNL2 and IFNL3 is being compared to the knockout of a single gene IFNL1. Rescue/add-back studies with single types at low/physiologal levels (whatever present in sup transfer) could further distinguish the activities of the subtypes if there are any.

Reviewer #3: 1. While the Ruxolitinib experiments are informative, they do not fully resolve the need for basal IFNλ2/3 signaling as the underlying mechanisms for increased viral susceptibility of IFNλ2/3 KO cells. Because the lines were single-cell cloned and already exhibit reduced STAT1/STAT2/IRF9 and blunted inducibility, clonal artifacts or loss of signaling components could explain the phenotype. I recommend the authors validate the phenotype in a pooled (polyclonal) edited population or in multiple independent KO clones and show that transient depletion (siRNA or neutralizing antibody) in parental cells reduces basal ISGs expression and conditioned-media activity to demonstrate that ongoing basal ligand signaling, rather than clone-specific effects or secondary loss of signaling machinery, is responsible for the observed phenotypes.

2. The authors have left the culture conditions undefined. Prior work shows that basal IFNλ2/3 induction is highly context-dependent (confluency, polarization, and Hippo-cGAS-STING signaling). Could the authors please clarify how the cells were cultured? Are the growth kinetics across clones equivalent? If not, is this a function of IFNλ signaling? Additional studies are needed to elucidate the expression levels of IFNλ1 and IFNλ2/3 under low and high confluency, as well as their induction during viral infection, to better understand the mechanisms that influence their expression, basal ISG expression, and the distinct role of IFNλ after viral challenge.

3. If cells can make both IFNλ1 and IFNλ2/3 during infection, and all three cytokines have similar activities in T84 cells, it is unclear why the authors propose that only IFNλ2/3 would confer protection against infection and basal ISG expression in T84 cells. Is this also the case in other intestinal epithelial cell lines and primary cells? Are these phenotypes conserved in other IFNλ-responsive epithelial cell types?

**Part III – Minor Issues: Editorial and Data Presentation Modifications**

Reviewer #1: • Lines 75-84: Would be appropriate to include findings from PMID 35137688 and 39893635 as part of this paragraph.

• Lines 100-102: It would also be appropriate to include PMID 31462571 as a prior study demonstrating a role for IFNL cytokines that is equivalent to the role for the receptor in mice.

• Is it possible to show the statistical comparisons between different doses and 0 in Figure 2? Perhaps with some sort of color scheme to demonstrate which comparisons are significantly different versus not?

• Comment further on the loss of viral susceptibility in IFNL1 KO cells in Fig 3?

• Fig 5A: Can the authors please clarify what is contributing to the large separation between IFNL2/3 and IFNLR KO lines? This is a somewhat surprising result.

• Fig 7: Were levels of the secreted basal IFNs (e.g. from WT cells) detectable by ELISA?

Reviewer #2: Minor comments

1. Line 44 “IFNL4 has been detrimental to humans' evolution…” is a pretty strong statement. It seems to have had some sort of benefit as it is highly represented in African populations (https://doi.org/10.1371/journal.pgen.1004681) so better to not make this conclusion that it is solely detrimental.

2. GSEA in supplementary figure 4 is not very convincing and has no pvalue. Not sure it adds anything to the genes shown in volcano plot and heatmap in fig 5.

3. Some of the text for RNAseq figures (5 and S5) is tiny and impossible to read. Consider reducing the number of labels or pathways shown and increasing fontsize

4. Line 414-417. Rodents have IFNL1 or an IFNL1-like gene but it is a pseudogene, so I don’t this this statement about its primate specificity is entirely accurate?

Reviewer #3: Supplementary figure 5: Consider plotting only significant terms to make the data more accessible.

PLOS authors have the option to publish the peer review history of their article (what does this mean? ). If published, this will include your full peer review and any attached files.

**Do you want your identity to be public for this peer review?** For information about this choice, including consent withdrawal, please see our Privacy Policy .

Reviewer #1: No

Reviewer #2: No

Reviewer #3: No

**Figure resubmission:**

**Reproducibility:**



---

## [Editor Report · Decision Letter 1]

29 Dec 2025

Dear Assistant Professor Stanifer,

We are pleased to inform you that your manuscript 'Basal IFNλ2/3 signaling is required for ISG expression and viral control in human intestinal epithelial cells' has been provisionally accepted for publication in PLOS Pathogens.

Best regards,

Helen M Lazear, Ph.D.

Academic Editor

PLOS Pathogens

Ashley St. John

Section Editor

PLOS Pathogens

Sumita Bhaduri-McIntosh

Editor-in-Chief

PLOS Pathogens

orcid.org/0000-0003-2946-9497

Michael Malim

Editor-in-Chief

PLOS Pathogens

orcid.org/0000-0002-7699-2064
---

## [Editor Report · Acceptance letter]

Dear Assistant Professor Stanifer,

We are delighted to inform you that your manuscript, "Basal IFNλ2/3 signaling is required for ISG expression and viral control in human intestinal epithelial cells," has been formally accepted for publication in PLOS Pathogens.

Best regards,

Sumita Bhaduri-McIntosh

Editor-in-Chief

PLOS Pathogens

orcid.org/0000-0003-2946-9497

Michael Malim

Editor-in-Chief

PLOS Pathogens

orcid.org/0000-0002-7699-2064